# *Memory Decoder*: A Pretrained, Plug-and-Play Memory for Large Language Models

**Jiaqi Cao**[1,4*]  **Jiarui Wang**[1*]  **Rubin Wei**[1]  **Qipeng Guo**[2]
**Kai Chen**[2]  **Bowen Zhou**[2,3]  **Zhouhan Lin**[1,2†]

[1]LUMIA Lab, School of Artificial Intelligence, Shanghai Jiao Tong University
[2]Shanghai AI Laboratory   [3]Tsinghua University
[4]SJTU Paris Elite Institute of Technology
maximus.cao@outlook.com, lin.zhouhan@gmail.com
code & model available at https://github.com/LUMIA-Group/MemoryDecoder

## Abstract

Large Language Models (LLMs) have shown strong abilities in general language tasks, yet adapting them to specific domains remains a challenge. Current method like Domain Adaptive Pretraining (DAPT) requires costly full-parameter training and suffers from catastrophic forgetting. Meanwhile, Retrieval-Augmented Generation (RAG) introduces substantial inference latency due to expensive nearest-neighbor searches and longer context. This paper introduces *Memory Decoder*, a plug-and-play pretrained memory that enables efficient domain adaptation without changing the original model's parameters. Memory Decoder employs a small transformer decoder that learns to imitate the behavior of an external non-parametric retriever. Once trained, Memory Decoder can be seamlessly integrated with any pretrained language model that shares the same tokenizer, requiring no model-specific modifications. Experimental results demonstrate that Memory Decoder enables effective adaptation of various Qwen and Llama models to three distinct specialized domains: biomedicine, finance, and law, reducing perplexity by an average of 6.17 points. Overall, Memory Decoder introduces a novel paradigm centered on a specially pretrained memory component designed for domain-specific adaptation. This memory architecture can be integrated in a plug-and-play manner, consistently enhancing performance across multiple models within the target domain.

## 1 Introduction

Large Language Models (LLMs) have demonstrated remarkable capabilities across a wide range of natural language processing tasks (Grattafiori et al., 2024; Yang et al., 2024; Liu et al., 2024; Guo et al., 2025). Pretrained on vast corpora of general text data, LLMs have revolutionized how we approach language understanding and generation tasks. However, despite their impressive general capabilities, adapting LLMs to perform optimally in specific domains remains a significant challenge. Domain-specific adaptation is crucial for applications in specialized fields such as biomedicine, finance, and law (Chen et al., 2023; Liu et al., 2023b; Colombo et al., 2024), where domain expertise and terminology are essential for accurate and reliable performance.

Domain adaptation for pretrained language models has traditionally followed several approaches, each with distinct advantages and limitations. Domain Adaptive Pre-Training (DAPT) involves continued pre-training of the LLM on domain-specific corpora (Gururangan et al., 2020).

---

*Equal contribution.
†Corresponding author.

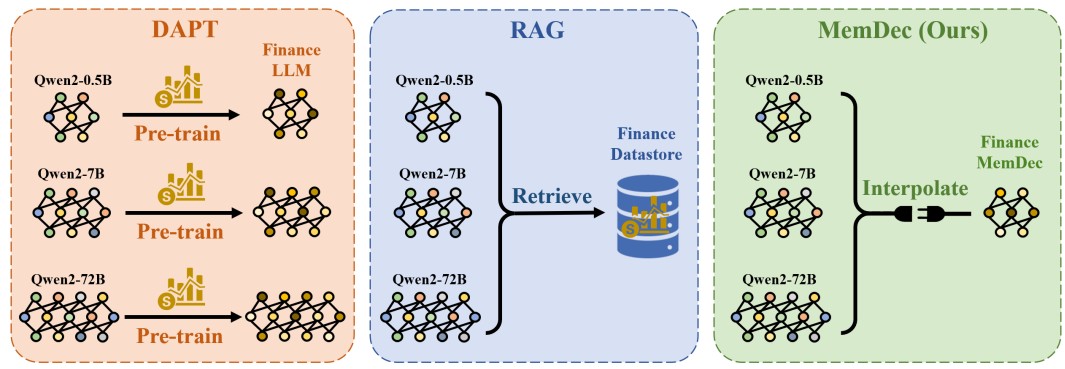

Figure 1: Comparison of domain adaptation approaches. DAPT (left) requires separate pre-training for each model size, modifying original parameters. RAG (middle) maintains model parameters but requires expensive retrieval from external datastores during inference. Memory Decoder (right) offers a plug-and-play solution where a single pretrained memory component can be interpolated with models of different sizes, avoiding both parameter modification and retrieval overhead.

While effective, this approach suffers from substantial computational costs associated with full-parameter training, especially as model sizes continue to grow into billions of parameters. Furthermore, adapting multiple models to the same domain requires separate training runs for each model, leading to resource inefficiency. Even with successful DAPT implementation, these models often encounter catastrophic forgetting, where the adaptation process diminishes the model's general capabilities (Kirkpatrick et al., 2017; Ven van de et al., 2024).

Retrieval-Augmented Generation (RAG) offers an alternative approach by enhancing model outputs with relevant retrieved information (Lewis et al., 2020; Izacard et al., 2023). While this method preserves the original model parameters, it introduces substantial computation overhead during inference due to expensive nearest neighbor (*kNN*) searches across large datastores and extended context (He et al., 2021).

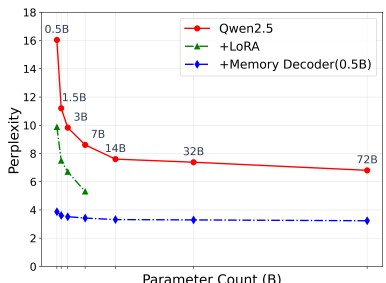

Figure 2: Perplexity comparison of Qwen2.5 models augmented by Memory Decoder and LoRA adapter of the same param count on the finance domain.

These two approaches present a fundamental dilemma in domain adaptation: DAPT requires costly training procedures and cannot efficiently adapt multiple models to the same domain, while RAG introduces significant computation and storage overhead during inference. This inherent trade-off between the plug-and-play nature of RAG and the inference efficiency of DAPT highlights the research gap for a solution offering both adaptability across models and computational efficiency during deployment. To address this challenge, we propose *Memory Decoder* (MemDec), a plug-and-play pretrained memory designed for efficient domain adaptation of large language models without modifying their parameters. Our approach draws inspiration from retrieval-based methods like kNN-LM (Khandelwal et al., 2019), but overcomes their limitations through a different paradigm. Rather than building and searching model-specific datastores during inference, Memory Decoder employs a small transformer decoder that is specially pretrained to **imitate the behavior of non-parametric retrievers** by aligning its output distribution with the ones of non-parametric retrievers. Figure 1 illustrates how our approach differs fundamentally from both DAPT and RAG.

The key innovation of our approach lies in its plug-and-play functionality: once trained, a single Memory Decoder can be seamlessly integrated with any large language model that shares the same tokenizer, without requiring model-specific adaptations or additional training. This architectural design enables immediate deployment across diverse model architectures, significantly reducing the computational resources needed for domain adaptation pre-training. Furthermore, unlike RAG methods, Memory Decoder achieves domain-specific performance improvements with minimal impact on inference latency, combining versatility with computational efficiency.

Experimental results across three specialized domains (biomedicine, finance, and law) and multiple model architectures demonstrate the versatility of Memory Decoder. As shown in Figure 2. the same Memory Decoder with only 0.5B parameters consistently enhances performance across seven different models from the Qwen2.5 model family on the finance domain. Our comprehensive analysis confirms that Memory Decoder successfully preserves the advantages of non-parametric approaches while eliminating their computational overhead, establishing a new paradigm for efficient domain adaptation of LLMs.

Our contributions can be summarized as follows:

- We introduce *Memory Decoder*, a plug-and-play pretrained memory that enables efficient domain adaptation for large language models without modifying their original parameters.
- We present the first approach that replaces traditional non-parametric retrievers with a compact parametric model, achieving superior performance while eliminating costly retrieval operations during inference.
- We demonstrate *Memory Decoder*'s generalizability, where a single domain-specific pretrained memory can be seamlessly integrated across all models with the same tokenizer.

## 2 Background

### 2.1 Problem Formulation

Domain adaptation aims to enhance a pretrained language model's performance on specialized text. Formally, given a pretrained model $\mathcal{M}_{\text{PLM}}$ with parameters $\theta$ and a domain corpus $\mathcal{D}_{\text{domain}}$, the goal is to optimize the next-token prediction distribution $p_{\text{PLM}}(y_t|x;\theta)$ for the target domain. Here, $x = (x_1, x_2, ..., x_{t-1})$ represents the context sequence and $y_t$ denotes the target token.

### 2.2 Nearest Neighbor Language Models

The $k$-nearest neighbor language model (kNN-LM) (Khandelwal et al., 2019) enables non-parametric domain adaptation without modifying the pretrained model's parameters.

For a domain corpus, kNN-LM first constructs a key-value datastore:

$$(K, V) = \{(\phi(x_i), y_i) \mid (x_i, y_i) \in \mathcal{D}_{\text{domain}}\} \tag{1}$$

where $\phi(\cdot)$ extracts hidden representations from the pretrained model.

During inference, for context $x$, it computes $k_t = \phi(x)$, retrieves $k$-nearest neighbors, and constructs a probability distribution:

$$p_{\text{kNN}}(y_t|x) \propto \sum_{(k_i, v_i) \in \mathcal{N}(k_t, k)} \mathbb{1}_{y_t = v_i} \exp(-d(k_t, k_i)/\tau) \tag{2}$$

The final prediction interpolates between the pretrained model and kNN distributions:

$$p_{\text{kNN-PLM}}(y_t|x) = \lambda \cdot p_{\text{kNN}}(y_t|x) + (1 - \lambda) \cdot p_{\text{PLM}}(y_t|x) \tag{3}$$

While effective, kNN-LM introduces substantial computational and storage overhead during inference. For instance, the Wikitext-103 datastore requires nearly 500GB storage even for GPT2-small model (He et al., 2021). These limitations motivate our Memory Decoder, a compact parametric model pretrained to mimic retrieval behavior while eliminating the need for large datastores.

## 3 Memory Decoder

In this section, we present Memory Decoder (MemDec), a plug-and-play pretrained memory designed for efficient domain adaptation of large language models. Our method consists of two primary components: a specialized pre-training procedure that aligns the output distribution of Memory Decoder with those of non-parametric retrievers (Section 3.1), and an efficient inference mechanism that enables plug-and-play domain adaptation (Section 3.2). As illustrated in Figure 3, Memory Decoder first learns to mimic non-parametric retrieval distributions during pre-training (upper part), then seamlessly integrates with any compatible language model during inference (lower part), eliminating the computational overhead associated with datastore maintenance and nearest neighbor search.

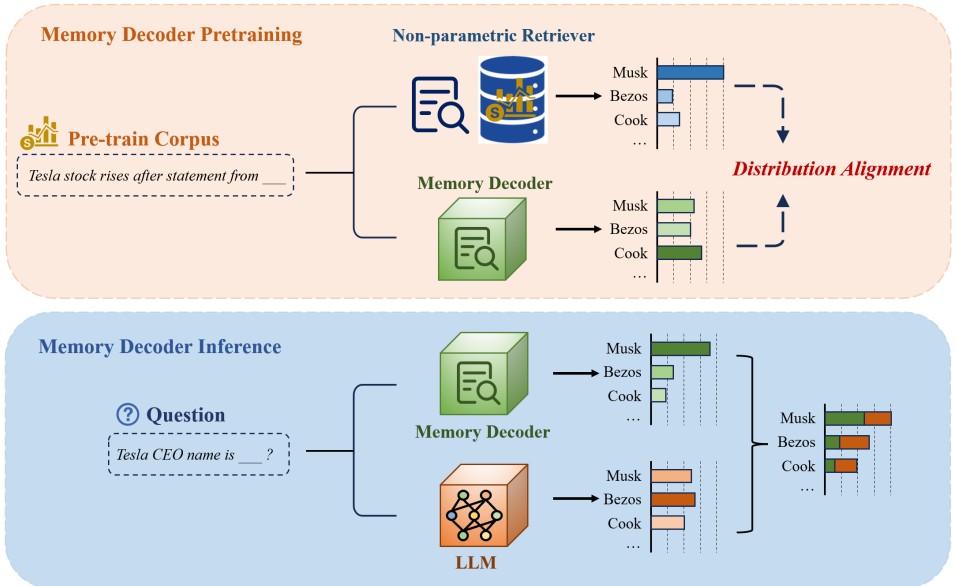

Figure 3: Overview of Memory Decoder architecture. **Upper**§ 3.1: During pre-training, Memory Decoder learns to align its output distributions with those generated by non-parametric retrievers through distribution alignment loss. **Lower**§ 3.2: During inference, Memory Decoder processes input in parallel with the base LLM, and their distributions are interpolated to produce domain-enhanced predictions without retrieval overhead.

## 3.1 Pre-training

Our primary goal during pre-training is to enable Memory Decoder $\mathcal{M}_{\text{Mem}}$ to produce probability distributions that closely resemble those generated by non-parametric retrievers when encountering the same context. This approach effectively encodes the domain knowledge captured in large key-value datastores into the parameters of our compact model.

**Data Construction**   Since we require non-parametric distributions as supervision signals, we construct training pairs of $(x_i, p_{\text{kNN}}(\cdot|x_i))$ in advance to enable efficient pre-training. Here, $x_i$ represents the input context and $p_{\text{kNN}}(\cdot|x_i)$ denotes the probability distribution generated by the non-parametric retriever for that context. First, we build a key-value datastore $(K, V) = \{(\phi(x_i), y_i) \mid (x_i, y_i) \in \mathcal{D}_{\text{train}}\}$ using our domain-specific corpus, where $\phi(\cdot)$ extracts hidden representations from a specific layer of the pretrained model. For each context $x_i$ in the corpus, we then perform $k$-nearest neighbor search against this datastore to identify similar contexts. To avoid trivial self-retrieval that would contaminate the learning signal, we exclude the top-1 neighbor where its key exactly matches the query key. Finally, we compute the non-parametric distribution $p_{\text{kNN}}(\cdot|x_i)$ for each context using the retrieved neighbors and cache these context-distribution pairs for training.

**Pre-training Objective**   Unlike traditional language modeling with single-label targets, kNN distributions offer richer supervision signals by capturing the diversity of plausible continuations in the domain (Xu et al., 2023)(see Appendix C for detailed analysis on kNN distributions). Through extensive experimentation, we have identified that a hybrid objective yields optimal performance.

Our approach centers on a Distribution Alignment Loss that minimizes the KL divergence (Van Erven, Harremos, 2014) between Memory Decoder's output distribution and the cached kNN distributions for each sample:

$$\mathcal{L}_{\text{KL}}(x_i) = \text{KL}(p_{\text{kNN}}(\cdot|x_i) \parallel p_{\text{Mem}}(\cdot|x_i)) \tag{4}$$

To prevent excessive deviation from the underlying corpus distribution, we integrate a complementary standard Language Modeling objective (Zhang, Sabuncu, 2018):

$$\mathcal{L}_{\text{LM}}(x_i) = -\log p_{\text{Mem}}(y_i|x_i) \tag{5}$$

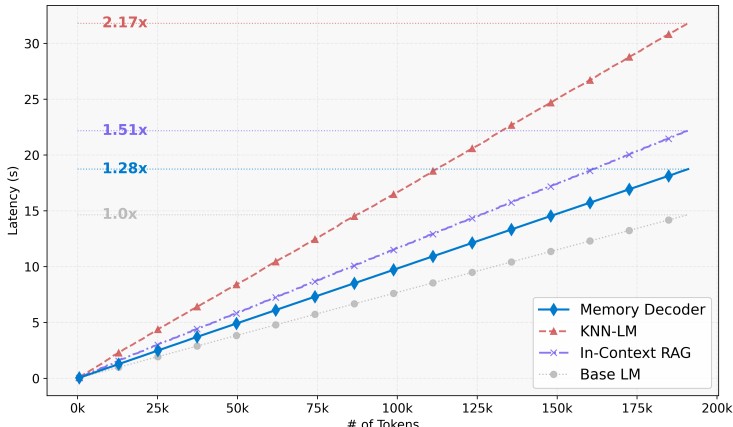

Figure 4: Inference latency comparison across domain adaptation methods. These measurements were conducted on Qwen2.5-1.5B (Yang et al., 2024) for biomedicine domain text, augmented by a 0.5B Memory Decoder.

The final loss function balances these two objectives through a hyperparameter $\beta$:

$$\mathcal{L}(x_i) = \beta \cdot \mathcal{L}_{\text{KL}}(x_i) + (1 - \beta) \cdot \mathcal{L}_{\text{LM}}(x_i) \tag{6}$$

Previous failed attempts to learn kNN distributions and our conjecture on why vanilla KL divergence with cross-entropy regularization succeeds are detailed in Appendix D.

## 3.2 Inference

Once pretrained, Memory Decoder exhibits a key plug-and-play capability that allows it to adapt any language model with a compatible tokenizer to the target domain via simple interpolation. During inference, both the pretrained language model $\mathcal{M}_{\text{PLM}}$ and Memory Decoder $\mathcal{M}_{\text{Mem}}$ process the same input context in parallel, and their output distributions are interpolated:

$$p_{\text{Mem-PLM}}(y_t|x) = \alpha \cdot p_{\text{Mem}}(y_t|x) + (1 - \alpha) \cdot p_{\text{PLM}}(y_t|x) \tag{7}$$

where $\alpha \in [0, 1]$ controls the influence of domain-specific knowledge.

Unlike traditional retrieval-augmented approaches that introduce substantial latency from nearest neighbor search and extended context processing, Memory Decoder requires only a single forward pass through a relatively small transformer decoder. As demonstrated in Figure 4, our method achieves significant improvements in inference efficiency compared to alternative domain adaptation techniques. With just 1.28× overhead relative to the base model, Memory Decoder substantially outperforms both In-Context RAG (Ram et al., 2023) (1.51×) and kNN-LM (Khandelwal et al., 2019) (2.17×). This computational advantage, combined with Memory Decoder's model-agnostic design, makes our approach particularly valuable for production environments where both performance and efficiency are critical considerations.

## 4 Experimental Setup

**Overview** We evaluate Memory Decoder across four complementary settings: (1) Language modeling on WikiText-103 (§5.1) to demonstrate effectiveness across GPT-2 model scales; (2) Downstream tasks (§5.2) to verify preservation of general capabilities during domain adaptation; (3) Cross-model adaptation (§5.3) showing a single Memory Decoder enhancing Qwen models from 0.5B to 72B parameters; (4) Cross-vocabulary adaptation (§5.4) demonstrating efficient transfer between tokenizer families; These experiments establish Memory Decoder as a versatile, plug-and-play solution for efficient domain adaptation across diverse architectures and applications.

**Datasets** For language modeling experiments, we use Wikitext-103 (Merity et al., 2016), a standard benchmark containing over 100M tokens from Wikipedia articles. For downstream evaluation,

|  | GPT2-small | GPT2-med | GPT2-large | GPT2-xl |
|---|---|---|---|---|
| **base** | 24.89 | 18.29 | 15.80 | 14.39 |
| *Non-parametric methods* | | | | |
| *+In-Context RAG* | 18.46 | 14.01 | 12.09 | 11.21 |
| *+kNN-LM* | 15.62 | 12.95 | 12.21 | 11.30 |
| *Parametric methods* | | | | |
| *+DAPT* | 14.76 | 12.78 | **11.10** | **10.16** |
| *+LoRA* | 18.63 | 13.88 | 11.77 | 10.67 |
| *+MemDec* | **13.36** | **12.25** | 11.53 | 10.93 |

Table 1: Perplexity comparison of domain adaptation methods across GPT2 model sizes on Wikitext-103. The best performing results are highlighted in **bold**, while the second-best results are underlined. Notably, applying our Memory Decoder(**124M**) to GPT2-medium(**345M**) outperforms DAPT of GPT2-medium(345M), demonstrating the effectiveness of our approach in capturing domain knowledge without modifying original parameters.

following the kNN-prompt framework, we assess performance across nine NLP tasks: sentiment analysis (SST2 (Socher et al., 2013), MR (Pang, Lee, 2005b), CR (Hu, Liu, 2004), RT (Pang, Lee, 2005a)), textual entailment (HYP (Kiesel et al., 2019), CB (De Marneffe et al., 2019), RTE (Dagan et al., 2010)), and text classification (AGN (Zhang et al., 2015a), Yahoo (Zhang et al., 2015b)). For domain-specific adaptation, we utilize three specialized corpora: (1) biomedical text from MIMIC-III (Johnson et al., 2016) clinical notes covering over 46,000 patients, (2) financial news (Liu et al., 2023a) from April 2024 to October 2024, and (3) legal text from the Asylex corpus (Barale et al., 2023) containing 59,112 documents of refugee status determination in Canada from 1996 to 2022.

**Baselines**  We compare Memory Decoder against several established domain adaptation methods: **In-Context RAG** (Ram et al., 2023), which implements a BM25 retriever that processes 32 query tokens, with retrieval occurring every 4 tokens. **kNN-LM** (Khandelwal et al., 2019), configured with interpolation parameter $\lambda = 0.25$ and temperature settings of $\tau = 1$ for GPT-2 small and medium, and $\tau = 13$ for large and xl models. **LoRA** (Hu et al., 2022), applied to query, key, value and MLP layers, with rank adjusted for each model to achieve parameter counts comparable to Memory Decoder. **Domain Adaptive Pretraining(DAPT)** (Gururangan et al., 2020), which involves complete retraining of all model parameters on the domain-specific corpus.

**Training Details**  We conduct our experiments on an 8×A800 80GB GPU setup. For language modeling and downstream evaluations, we use a GPT2-xl model(finetuned on wikitext) to build the key-value datastore and non-parametric distributions for training, and continue training on a GPT2-small model(finetuned on wikitext) with learning rate 1e-3. For cross-model adaptation, we use Qwen2.5-1.5B (Yang et al., 2024) to build the datastore, and continue training on Qwen2.5-0.5B with learning rate 1e-4. For cross-vocabulary adaptation, we use Llama3.2-1B (Grattafiori et al., 2024) to build the datastore, and continue training on the Memory Decoder trained from cross-model experiments, with its embedding layer and language model head re-initialized. All experiments use a training budget equivalent to the computational cost of training a 7B parameter model for 1 epoch, with DAPT and LoRA baselines using the same maximum training FLOPS but early stopped to prevent overfitting. The training hyperparameter $\beta$ is set to 0.5 across all tasks.

**Evaluation Metrics**  For language modeling, cross-model, and cross-tokenizer experiments, we use sliding window perplexity. Following Baevski, Auli (2018), in each test example, the context length is set to 1024 where only the latter 512 tokens are scored. For downstream evaluation, following methodology from Shi et al. (2022), we report results using the domain-conditional PMI scoring rule (Holtzman et al., 2021). The interpolation hyperparameter $\alpha$ is tuned on the validation split of each task following Khandelwal et al. (2019), see more details in Appendix A.

|  | SST2 | MR | CR | RT | HYP | CB | RTE | AGN | Yahoo | Avg |
|---|---|---|---|---|---|---|---|---|---|---|
| **base** | 81.98 | 78.40 | 84.40 | 76.54 | 63.75 | 41.07 | 52.70 | 78.79 | 49.40 | 67.45 |
| *Non-parametric methods* | | | | | | | | | | |
| *+kNN-LM* | 81.98 | 77.95 | 83.80 | 77.95 | 64.14 | 39.28 | 52.70 | 77.73 | 49.63 | 67.24 |
| *Parametric methods* | | | | | | | | | | |
| *+DAPT* | 83.52 | 80.15 | 80.45 | 77.39 | 36.04 | 50.00 | 51.26 | 64.31 | 24.40 | 60.84 |
| *+LoRA* | 80.88 | 76.90 | 83.95 | 76.07 | 64.14 | 39.28 | 53.79 | 81.06 | 49.46 | 67.28 |
| *+MemDec* | 82.43 | 78.35 | 84.35 | 77.30 | 64.15 | 57.14 | 55.24 | 79.80 | 49.37 | **69.79** |

Table 2: Performance on nine diverse NLP tasks including sentiment analysis, textual entailment, and text classification.

## 5 Results

### 5.1 Language Modeling on Wikitext-103

Table 1 demonstrates the exceptional effectiveness of Memory Decoder across all GPT2 model sizes. A single Memory Decoder with only 124M parameters consistently enhances the entire GPT2 family, showcasing its plug-and-play capability regardless of base model size. For smaller models, our approach delivers superior results compared to all adaptation methods—notably maintaining an advantage for GPT2-medium despite utilizing only one third of the parameters. Even when applied to larger models where DAPT has inherent advantages due to full model updates, Memory Decoder remains highly competitive while consistently outperforming all other parameter-efficient methods without modifying any original parameters. These results validate that a small parametric decoder can effectively capture the benefits of non-parametric retrieval while eliminating computational overhead.

### 5.2 Downstream Performance

Table 2 reveals Memory Decoder's ability to enhance domain adaptation while preserving general language capabilities in zero-shot evaluation settings. Unlike DAPT, which suffers catastrophic forgetting on several tasks (particularly HYP and Yahoo where performance drops by nearly half; see Appendix B for detailed analysis), Memory Decoder maintains or improves performance across all evaluated tasks. Our approach achieves the highest average score across all nine tasks, outperforming the base model, kNN-LM, and LoRA while demonstrating particular strength on textual entailment tasks like CB and RTE. These results validate a key advantage of our plug-and-play architecture: by keeping the original model parameters intact while augmenting them with domain knowledge, Memory Decoder achieves domain adaptation without sacrificing general capabilities. Importantly, all experiments are conducted in a zero-shot setting, and our method should be viewed as orthogonal to in-context learning approaches.

### 5.3 Cross-Model Adaptation

Table 3 demonstrates Memory Decoder's exceptional plug-and-play capabilities across diverse model sizes and architectures. A single Memory Decoder (0.5B parameters) consistently enhances performance across all models in both the Qwen2 and Qwen2.5 families, spanning from 0.5B to 72B parameters. For smaller models like Qwen2-0.5B, our approach achieves dramatic perplexity reductions—transforming domain-specific performance to state-of-the-art results on both biomedical and financial text. Even for the largest models in the family, Memory Decoder provides substantial improvements, demonstrating that retrieval-augmented knowledge remains valuable regardless of model scale. These results validate Memory Decoder's core strength: a single pretrained memory component can enhance multiple models sharing the same tokenizer, providing efficient domain adaptation that scales from the smallest to the largest models while consistently outperforming existing approaches.

### 5.4 Cross-Vocabulary Adaptation

Table 4 demonstrates Memory Decoder's ability to generalize across different tokenizers and model architectures. By re-initializing only the embedding layer and language model head of our Qwen2.5-trained Memory Decoder, we successfully adapt it to the Llama model family with just 10% of the original training budget. This efficient transfer enables substantial performance improvements across

| Model | Bio | Fin | Law | Avg |
|---|---|---|---|---|
| *Qwen2 Family* | | | | |
| **Qwen2-0.5B** | 18.41 | 16.00 | 10.23 | 14.88 |
| *+LoRA* | 7.28 | 9.70 | 5.82 | 7.60 |
| *+MemDec* | **3.75** | **3.84** | **4.57** | **4.05** |
| **Qwen2-1.5B** | 12.42 | 10.96 | 7.69 | 10.36 |
| *+LoRA* | 5.73 | 7.37 | 4.84 | 5.98 |
| *+MemDec* | **3.68** | **3.61** | **4.32** | **3.87** |
| **Qwen2-7B** | 8.36 | 8.31 | 5.92 | 7.53 |
| *+LoRA* | 4.47 | 5.64 | 4.02 | 4.71 |
| *+MemDec* | **3.59** | **3.38** | **4.00** | **3.66** |
| **Qwen2-72B** | 6.15 | 6.62 | 4.84 | 5.87 |
| *+MemDec* | **3.45** | **3.20** | **3.69** | **3.45** |
| *Qwen2.5 Family* | | | | |
| **Qwen2.5-0.5B** | 17.01 | 16.04 | 9.86 | 14.30 |
| *+LoRA* | 7.02 | 9.88 | 5.75 | 7.55 |
| *+MemDec* | **3.74** | **3.87** | **4.57** | **4.06** |
| **Qwen2.5-1.5B** | 11.33 | 11.20 | 7.42 | 9.98 |
| *+LoRA* | 5.59 | 7.50 | 4.82 | 5.97 |
| *+MemDec* | **3.67** | **3.61** | **4.29** | **3.86** |
| **Qwen2.5-3B** | 9.70 | 9.83 | 6.68 | 8.74 |
| *+LoRA* | 5.07 | 6.71 | 4.45 | 5.41 |
| *+MemDec* | **3.63** | **3.52** | **4.16** | **3.77** |
| **Qwen2.5-7B** | 8.19 | 8.61 | 5.94 | 7.58 |
| *+LoRA* | 4.03 | 5.31 | **3.81** | 4.38 |
| *+MemDec* | **3.57** | **3.42** | 4.01 | **3.67** |
| **Qwen2.5-14B** | 7.01 | 7.60 | 5.35 | 6.65 |
| *+MemDec* | **3.51** | **3.31** | **3.86** | **3.56** |
| **Qwen2.5-32B** | 6.65 | 7.38 | 5.18 | 6.40 |
| *+MemDec* | **3.48** | **3.29** | **3.81** | **3.53** |
| **Qwen2.5-72B** | 5.90 | 6.80 | 4.84 | 5.85 |
| *+MemDec* | **3.44** | **3.23** | **3.70** | **3.46** |

Table 3: Cross-model adaptation results across three specialized domains. A single 0.5B Memory Decoder enhances models ranging from 0.5B to 72B parameters.

| Model | Bio | Fin | Law | Avg |
|---|---|---|---|---|
| *Llama3 Family* | | | | |
| **Llama3-8B** | 7.95 | 8.63 | 5.96 | 7.51 |
| *+LoRA* | 4.38 | 5.68 | **4.12** | 4.73 |
| *+MemDec* | **3.92** | **4.32** | 4.46 | **4.23** |
| **Llama3-70B** | 5.92 | 6.87 | 4.90 | 5.90 |
| *+MemDec* | **3.74** | **4.01** | **4.07** | **3.94** |
| *Llama3.1 Family* | | | | |
| **Llama3.1-8B** | 7.82 | 8.46 | 5.88 | 7.39 |
| *+LoRA* | 4.38 | 5.72 | **4.10** | 4.73 |
| *+MemDec* | **3.91** | **4.30** | 4.42 | **4.21** |
| **Llama3.1-70B** | 5.85 | 6.68 | 4.89 | 5.81 |
| *+MemDec* | **3.73** | **3.97** | **4.06** | **3.92** |
| *Llama3.2 Family* | | | | |
| **Llama3.2-1B** | 12.81 | 11.85 | 8.23 | 10.96 |
| *+LoRA* | 5.97 | 7.83 | 5.21 | 6.34 |
| *+MemDec* | **4.06** | **4.85** | **5.11** | **4.67** |
| **Llama3.2-3B** | 9.83 | 9.70 | 6.83 | 8.79 |
| *+LoRA* | 5.11 | 6.55 | **4.59** | 5.42 |
| *+MemDec* | **3.99** | **4.45** | 4.76 | **4.40** |

Table 4: Cross-vocabulary adaptation results demonstrating efficient knowledge transfer between model families. Memory Decoder trained on Qwen2.5 can be adapted to Llama models with minimal additional training (10% of original budget), achieving substantial perplexity reductions across all Llama variants and consistently outperforming LoRA in biomedical and financial domains.

all Llama variants. For Llama3-8B, Memory Decoder achieves roughly 50% perplexity reduction on both biomedical and financial domains. Similar improvements extend to the Llama3.1 and Llama3.2 families, with our method consistently outperforming LoRA on biomedical and financial domains, though showing room for improvement on legal text. These findings illustrate Memory Decoder's versatility beyond a single tokenizer family, demonstrating that domain knowledge learned from one architecture can be efficiently transferred to another with minimal additional training. This capability expands the practical utility of our approach, offering a streamlined path to domain adaptation across diverse model ecosystems.

# 6 Analysis

## 6.1 Case Study: Bridging Parametric and Non-Parametric Methods

Memory Decoder fundamentally learns to compress the knowledge stored in large non-parametric datastores into a compact parametric model, combining the memorization capabilities of retrieval methods with the efficiency and generalization of parametric approaches. To validate this hypothesis, we conducted case studies on WikiText-103 examining how different methods assign probabilities to specific tokens.

As shown in Table 5, Memory Decoder exhibits two crucial capabilities:

**Long-tail Knowledge:** For factual information like "Jacobi" and "1906", Memory Decoder assigns dramatically higher probabilities than the base model (68.94% vs. 0.12% and 98.65% vs. 1.57%), successfully capturing the memorization benefits of non-parametric methods.

**Semantic Coherence:** For function words and logical continuations like "on" and "C", Memory Decoder maintains probabilities closer to the base model rather than following kNN-LM's lower probabilities, demonstrating its ability to preserve coherent language modeling capabilities that pure retrieval methods sacrifice.

These observations confirm that Memory Decoder occupies a unique position: it enhances memorization of domain-specific and long-tail knowledge like non-parametric methods, while maintaining the generalization and reasoning capabilities inherent to parametric models.

| Long-tail Knowledge Learning | | | |
|---|---|---|---|
| **Context (target token underlined)** | **MemDec** | **kNN** | **Base LM** |
| he starred alongside actors Mark Strong and Derek Jacobi | **68.94%** | 9.39% | 0.12% |
| The launch of HMS Dreadnought in 1906 by the Royal Navy raised the stakes | **98.65%** | 40.62% | 1.57% |
| Semantic Coherence and Reasoning | | | |
| **Context (target token underlined)** | **MemDec** | **kNN** | **Base LM** |
| In 2000 Boulter had a guest-starring role on the television series The Bill | 40.11% | 8.07% | **45.51%** |
| ...three tank squadrons for special overseas operations, known as 'A', 'B' and ' C ' Special Service Squadrons | 50.10% | 10.76% | **63.04%** |

Table 5: **Probability assignments for specific tokens by different methods.** *Orange section* : Memory Decoder excels at capturing long-tail factual knowledge, assigning dramatically higher probabilities than the base model. *Cyan section* : For semantic coherence, Memory Decoder intelligently balances between kNN-LM and base model probabilities, preserving linguistic fluency.

| | GPT2-small | GPT2-medium | GPT2-large | GPT2-xl | Avg |
|---|---|---|---|---|---|
| Base | 24.89 | 18.29 | 15.80 | 14.39 | 18.34 |
| DAPT | 14.76 | 12.78 | 11.10 | 10.16 | 12.20 |
| *+ MemDec-small (124M)* | 13.36 | 12.25 | 11.53 | 10.93 | 12.01 |
| *+ MemDec-medium (345M)* | 12.08 | 11.59 | 10.92 | 10.43 | 11.26 |
| *+ MemDec-large (774M)* | **11.67** | **11.23** | **10.83** | **10.28** | **11.00** |

Table 6: Performance comparison with different Memory Decoder sizes. Even small Memory Decoders achieve competitive performance with full-parameter DAPT while maintaining plug-and-play capability.

## 6.2 Impact of Memory Decoder Size

Table 6 examines how Memory Decoder size affects performance across the GPT2 family. As Memory Decoder size increases, performance consistently improves across all base models, with the large variant achieving the best average perplexity. These results validate that Memory Decoder provides an efficient alternative to full model fine-tuning: practitioners can choose the decoder size based on their computational constraints while maintaining the crucial advantage of preserving the original model's capabilities.

## 6.3 Ablation on the pre-training objective

We compare Memory Decoder against logit interpolation with a DAPT model. Table 7 shows Memory Decoder consistently outperforms DAPT interpolation across all GPT2 scales, with an average gain of 1.90 perplexity points. Notably, the gains persist even at larger scales, confirming that our hybrid pre-training objective provides complementary value beyond what standard language modeling objectives can achieve, regardless of model size.

## 7 Related Work

**Retrieval-Augmented Generation**  Retrieval-Augmented Generation (RAG) enhances language models by incorporating knowledge from external sources, with retrieval granularity ranging from documents (Chen et al., 2017) to passages (Guu et al., 2020; Lewis et al., 2020; Izacard et al., 2023) to tokens (Khandelwal et al., 2019; He et al., 2021; Min et al., 2022; Yogatama et al., 2021). Token-level retrieval achieves superior performance for rare patterns and out-of-domain scenarios but introduces substantial computation overhead during inference. While non-differentiable retrieval mechanisms prevent end-to-end optimization and memory token approaches (Chevalier et al., 2023) enable differentiable access but are limited to local contexts, Memory Decoder provides both differentiable

| Base Model | Baseline PPL | + DAPT-small | + MemDec-small |
|---|---|---|---|
| GPT2-small | 24.89 | 15.95 | **13.36** (-2.59) |
| GPT2-medium | 18.29 | 14.26 | **12.25** (-2.01) |
| GPT2-large | 15.80 | 13.13 | **11.53** (-1.60) |
| GPT2-xl | 14.39 | 12.30 | **10.93** (-1.37) |
| **Average** | 18.34 | 13.91 | **12.01** (-1.90) |

Table 7: Memory Decoder vs. DAPT model interpolation on WikiText-103. Both use 124M parameter models with different training objectives.

optimization and full-dataset knowledge access without expensive retrieval operations or model-specific datastores.

**Domain Adaptation**   Domain adaptation techniques have evolved from domain-specific pre-training (SciBERT (Beltagy et al., 2019), BioBERT (Lee et al., 2020), ClinicalBERT (Huang et al., 2019)) to parameter-efficient methods like LoRA (Hu et al., 2022) and adapters (Wang et al., 2020; Diao et al., 2021, 2023). However, these approaches require model-specific modifications, preventing generalization across architectures. Memory Decoder addresses this limitation by providing a domain-specific memory module that enhances multiple frozen language models without parameter modifications, enabling cross-model adaptation within tokenizer families and efficient cross-tokenizer transfer with minimal additional training.

# 8    Conclusion

In this paper, we introduced Memory Decoder, a novel plug-and-play approach for domain adaptation of large language models. By pre-training a small transformer decoder to emulate the behavior of non-parametric retrievers, Memory Decoder effectively adapts any compatible language model to a specific domain without modifying its parameters. Our comprehensive experiments across multiple model families and specialized domains demonstrate that Memory Decoder consistently outperforms both parametric adaptation methods and traditional retrieval-augmented approaches.

The key innovation of Memory Decoder lies in its versatility and efficiency. A single pretrained Memory Decoder can seamlessly enhance any model that shares the same tokenizer, and with minimal additional training, can be adapted to models with different tokenizers and architectures. This capability enables efficient domain adaptation across model families, dramatically reducing the resources typically required for specialized model development. Our results confirm that Memory Decoder preserves the performance benefits of retrieval-augmented methods while maintaining the general capabilities of the base models, avoiding the catastrophic forgetting often observed with fine-tuning approaches.

Memory Decoder introduces a new paradigm for domain adaptation that fundamentally reimagines how we specialize language models for particular domains. By decoupling domain expertise from model architecture through a pretrained memory component, our approach creates a more modular, efficient, and accessible framework for enhancing language model performance in specialized fields.

# 9    Limitations

While Memory Decoder demonstrates significant advantages for domain adaptation, we acknowledge several limitations in our current approach. The pre-training phase requires searching in key-value datastores to obtain kNN distributions as training signals, introducing computational overhead during the training process. Although this cost is incurred only once per domain and is amortized across all adapted models, it remains a bottleneck in the pipeline. Additionally, while cross-tokenizer adaptation requires minimal training compared to training from scratch, it still necessitates some parameter updates to align embedding spaces, preventing truly zero-shot cross-architecture transfer.

## Acknowledgement

This work is sponsored by the National Key Research and Development Program of China (No. 2023ZD0121402), Shanghai Fundamental Research Program for General AI Models (No. 2025SHZDZX0251101), and the National Natural Science Foundation of China (NSFC) grant (No.62576211).

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

# A  Interpolation hyperparameter $\alpha$ of all tasks

## A.1  Language Modeling on Wikitext-103

For language modeling on WikiText-103 (section 5.1), we use the following $\alpha$ values for different GPT-2 model sizes:

| Model | $\alpha$ |
|---|---|
| GPT-2-small | 0.80 |
| GPT-2-medium | 0.60 |
| GPT-2-large | 0.55 |
| GPT-2-xl | 0.55 |

Table 8: Interpolation hyperparameter $\alpha$ for GPT-2 models on WikiText-103.

The trend of smaller $\alpha$ for larger GPT models aligns with intuition—stronger base models require less augmentation from the memory component. The general pattern centers around $\alpha$=0.6, confirming it as a robust default choice.

## A.2  Downstream Performance

Table 9 presents the optimal $\alpha$ values for downstream tasks in section 5.2.

| Task | $\alpha$ |
|---|---|
| SST-2 | 0.30 |
| MR | 0.30 |
| CR | 0.05 |
| RT | 0.20 |
| HYP | 0.20 |
| CB | 0.30 |
| RTE | 0.60 |
| AGN | 0.20 |
| Yahoo | 0.20 |

Table 9: Optimal interpolation hyperparameter $\alpha$ for downstream tasks.

The general pattern centers around $\alpha$=0.3, which is consistent with the findings in Shi et al. (2022).

## A.3  Cross-Model and Cross-Vocabulary Adaptation

For domain-specific language modeling tasks (section 5.3 and 5.4), we tune $\alpha$ on the validation set by searching over $\{0.4, 0.6, 0.8, 0.9\}$.

# B  Analysis of DAPT Performance on Downstream Tasks

Previous work has shown that domain-adaptive pretraining can adversely affect a model's prompting ability (Cheng et al., 2023). Our experiments reveal that this effect is particularly pronounced when using domain-conditional PMI (DCPMI) scoring for evaluation, especially on tasks where label verbalizers overlap with the pretraining domain vocabulary.

As shown in Table 10, while direct language modeling evaluation reveals only modest performance drops with DAPT, the DCPMI scores show dramatic degradation for smaller models. This discrepancy arises because we employ fuzzy verbalizers following Shi et al. (2022), and the label spaces for Yahoo and AGN tasks contain terms (e.g., "politics," "technology") that appear frequently in WikiText-103. When DAPT increases the domain probability for these terms, it causes the conditional PMI scores to drop substantially, as the denominator in the DCPMI calculation becomes inflated.

The results for GPT-2-xl demonstrate that larger models exhibit greater robustness to this evaluation artifact, maintaining relatively stable DCPMI scores after domain adaptation. This suggests that

| Model | Yahoo (LM) | Yahoo (DCPMI) | HYP (LM) | HYP (DCPMI) | Avg |
|---|---|---|---|---|---|
| GPT-2-small | 0.466 | 0.495 | 0.639 | 0.638 | 0.559 |
| +DAPT | 0.429 | 0.244 | 0.608 | 0.361 | 0.410 |
| Δ | -0.037 | -0.251 | -0.031 | -0.277 | -0.149 |
| GPT-2-xl | 0.520 | 0.499 | 0.628 | 0.609 | 0.564 |
| +DAPT | 0.490 | 0.491 | 0.624 | 0.618 | 0.556 |
| Δ | -0.030 | -0.008 | -0.004 | +0.009 | -0.008 |

Table 10: Comparison of standard language modeling (LM) scores versus domain-conditional PMI (DCPMI) scores for DAPT models on Yahoo and HYP tasks.

the apparent failure of DAPT on certain downstream tasks is partly an artifact of the evaluation methodology rather than a fundamental limitation of the approach, though the phenomenon highlights an important interaction between domain adaptation and prompt-based evaluation methods.

## C   Characteristics of $k$-NN Distributions

### C.1   Extreme Sparsity and Concentration

$k$-NN distributions exhibit fundamentally different characteristics from standard language model outputs. While LM distributions maintain smooth probability mass across vocabulary with extensive long tails, $k$-NN distributions demonstrate extreme sparsity—typically assigning non-zero probabilities to only 2–3 tokens from a 50,257-dimensional vocabulary.

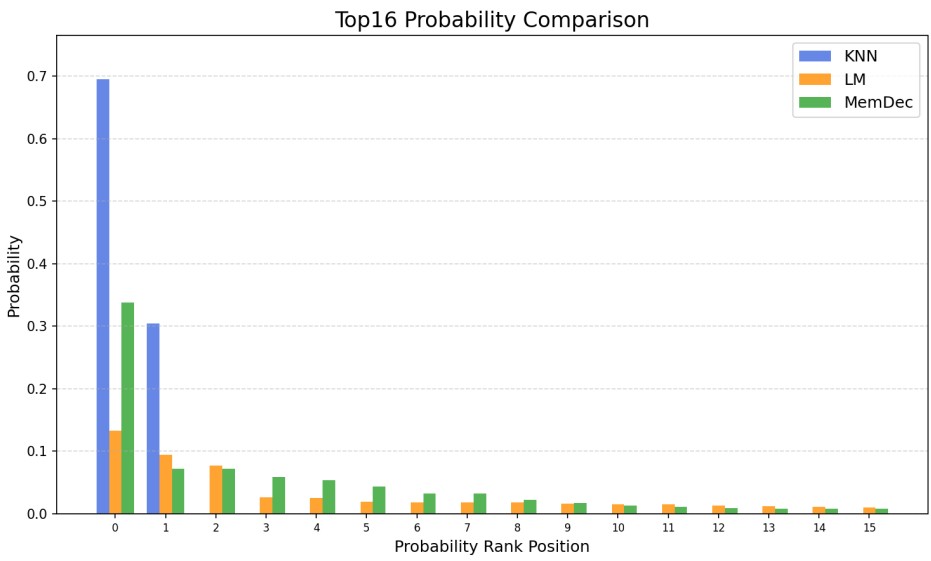

Figure 5: Probability distributions from $k$-NN retrieval, standard LM, and Memory Decoder for GPT-2-Large. The $k$-NN distribution shows extreme sparsity with concentrated probability mass.

This concentration emerges from two factors: (1) the hard constraint of selecting only $k$ nearest neighbors eliminates low-probability candidates, and (2) high-dimensional embedding spaces (e.g., 1280 dimensions for GPT-2-Large) amplify distance relationships through the curse of dimensionality, causing nearest neighbors to dominate disproportionately.

### C.2   Scale-Dependent Behavior

Model scale dramatically affects $k$-NN distribution quality. GPT-2-small (124M) produces distributions marginally different from its LM outputs (top-1 probability 50%), while GPT-2-Large

(1.5B) generates radically sparse distributions with 93.48% average top-1 probability—a 67% relative increase over its baseline.

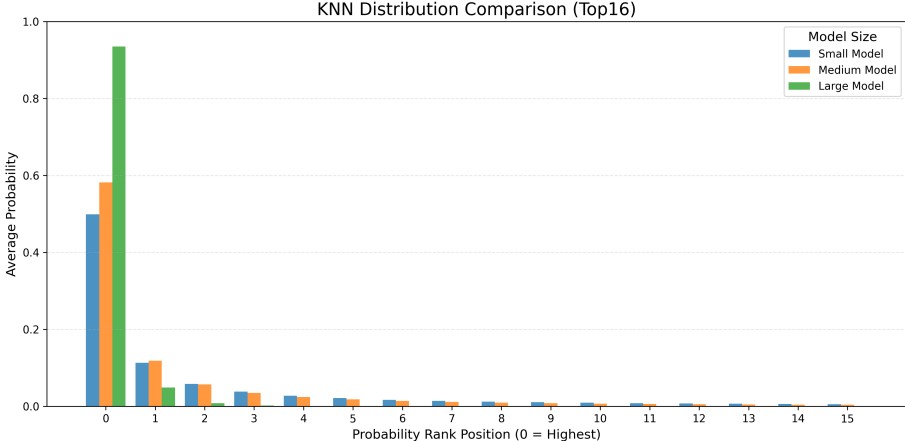

Figure 6: $k$-NN distribution sparsity across model scales. Despite identical retrieval parameters ($k = 1024$), larger models produce substantially sparser distributions.

| Base Model | Baseline | PPL with MemDec from: | | |
|---|---|---|---|---|
| | PPL | Small | Medium | Large |
| GPT-2-Small | 24.89 | 14.01 | 13.80 | **13.77** |
| GPT-2-Medium | 18.29 | 12.88 | 12.74 | **12.70** |
| GPT-2-Large | 15.80 | 12.05 | 11.95 | **11.93** |

Table 11: Perplexity with Memory Decoder (125M) trained using $k$-NN distributions from different source models

Larger models benefit from: (1) higher-dimensional spaces where distance concentration intensifies, and (2) superior contextual representations that better disambiguate polysemous tokens and preserve semantic distinctions, leading to more coherent nearest neighbor retrievals.

### C.3 Domain Adaptation Effects

Fine-tuned models produce sharper $k$-NN distributions than base models. Domain adaptation creates specialized embedding clusters with reduced intra-cluster variance and increased inter-cluster separation, leading to more decisive retrievals. Memory Decoders trained with fine-tuned distributions consistently achieve lower perplexity, validating that domain-adapted representations provide superior retrieval targets.

## D  Alternative Loss Functions for Imitating $k$-NN Distributions

### D.1  Failed Approaches

While KL divergence (combined with cross-entropy regularization) successfully matches $k$-NN distributions, we systematically evaluated several alternative loss functions that all demonstrated inferior performance:

#### D.1.1  Focal Loss

Adapted from object detection to handle class imbalance through gradient rescaling:

$$\mathcal{L}_{\text{Focal}} = -\sum_i \left[ \alpha(1 - p_\theta(i))^\gamma p_{\text{kNN}}(i) \log p_\theta(i) + (1 - \alpha)p_\theta(i)^\gamma (1 - p_{\text{kNN}}(i)) \log(1 - p_\theta(i)) \right]$$

$$(8)$$

With $\alpha = 0.5$ and $\gamma = 2$, focal loss theoretically emphasizes hard-to-classify sparse regions but failed to achieve sufficient distribution concentration in practice.

### D.1.2 Jensen-Shannon Divergence

A symmetric alternative to KL divergence:

$$\text{JSD}(P \parallel Q) = \frac{1}{2}D_{\text{KL}}(P \parallel M) + \frac{1}{2}D_{\text{KL}}(Q \parallel M), \quad M = \frac{1}{2}(P + Q) \tag{9}$$

Despite avoiding the directional bias of KL divergence, JSD provided no advantage for our extremely sparse target distributions.

### D.1.3 Bi-directional Logits Difference (BiLD)

BiLD focuses on relative rankings by computing pairwise differences among top-$k$ logits:

$$\mathcal{L}_{\text{BiLD}} = D_{\text{KL}}[p_{\text{led}}^{\text{kNN}} \| p_{\text{cor}}^{\theta}] + D_{\text{KL}}[p_{\text{cor}}^{\text{kNN}} \| p_{\text{led}}^{\theta}] \tag{10}$$

While theoretically suited for distributions where relative ordering matters more than exact probabilities, BiLD consistently underperformed standard KL divergence.

### D.1.4 Explicit Sparsity Penalty

Direct penalization of non-zero predictions in zero-probability regions:

$$\mathcal{L}_{\text{sparse}} = \mathcal{L}_{\text{KL}} + \alpha \sum_i \mathbb{I}_{\{p_{\text{kNN}}(i)=0\}} \cdot p_\theta(i), \quad \alpha = 0.01 \tag{11}$$

This approach created training instability without meaningfully improving output sparsity.

## D.2 Why KL Divergence Succeeds

The superior performance of KL divergence (with cross-entropy regularization) for matching $k$-NN distributions likely stems from its unique mathematical properties that align with the retrieval-based nature of the target:

**Asymmetric penalty structure**: KL divergence $D_{\text{KL}}(P\|Q) = \sum_i P(i) \log \frac{P(i)}{Q(i)}$ heavily penalizes placing probability mass where the target has none (when $P(i) \approx 0$ but $Q(i) > 0$), while being more forgiving of missing mass where the target has some. This asymmetry naturally encourages sparsity—the model learns to aggressively zero out predictions outside the $k$-NN support.

**Mode-seeking behavior**: The forward KL divergence $D_{\text{KL}}(P\|Q)$ is inherently mode-seeking, preferring to capture a few high-probability modes rather than covering the entire distribution. For $k$-NN distributions with 2-3 dominant modes, this bias perfectly matches the desired behavior, unlike symmetric losses (JSD) or mode-covering alternatives.

**Information-theoretic optimality**: KL divergence directly minimizes the expected encoding length difference between distributions. For $k$-NN distributions that encode "retrieval-aware uncertainty," KL naturally preserves the information structure—maintaining both the sharp peaks (high retrieval confidence) and the specific ranking among top candidates that emerges from the datastore's empirical distribution.

The cross-entropy regularization component anchors the model to linguistically valid outputs, preventing collapse to degenerate solutions while the KL term drives sparsity. This combination uniquely balances the competing demands of extreme concentration and semantic coherence.

