# OpenReview forum: "Memory Decoder: A Pretrained, Plug-and-Play Memory for Large Language Models"
_NeurIPS.cc/2025/Conference — NeurIPS 2025 poster_

### Official Review · Reviewer_2Vi7 · 2025-07-01

**Clarity:** 3
**Significance:** 3
**Originality:** 3
**Rating:** 4
**Confidence:** 4

**Summary:**

This paper proposes a memory decoder, an LM that is trained to mimic a nonparametric LM (specifically, kNN-LM) and then augmented with a standard LM at inference time. The idea is that for domain adaptation, continued training (or fine-tuning) leads to catastrophic forgetting, whereas standard retrieval-augmented LMs or nonparametric LMs increase the latency. The memory decoder aims to address these problems and benefit from both worlds. Experiments are done in various evaluation settings including: a standard perplexity eval, downstream task fine-tuning, and domain adaptation. Results show that the proposed method outperforms baselines including retrieval-augmented LMs, kNN-LM, full fine-tuning and LoRA tuning, while having very little computational/latency overhead.

**Questions:**

Written in the previous field

**Ethical Concerns:**

["NO or VERY MINOR ethics concerns only"]

**Final Justification:**

Revised after rebuttal

**Quality:**

3

**Strengths And Weaknesses:**

Strengths
- The problem is well motivated – the limitations of standard training and retrieval augmentation are well explained, and the memory decoder is shown to be effective in achieving the best of both worlds.
- Experiments are comprehensive - although it is on a small scale and evaluated on relatively easy datasets by today’s standard, it is quite comprehensive and includes a thorough set of baselines.
- All the execution steps are all reasonable and sound.
- The paper is well written and is easy to follow.

Weaknesses/questions
- A fundamental question that arises to me after reading this paper is – a nonparametric LM and finetuning of LM have different properties, rather than one being universally better than the other, e.g., a nonparametric model may be better at long-tail or uncommon patterns like factoid knowledge, while finetuning of an LM may make the model generalize better. Is the proposed model closer to a nonparametric LM, or to finetuning of LM, in terms of model properties? I think this is a fundamental question that should be discussed and analyzed.
- The paper claims that  the memory decoder prevents catastrophic forgetting that standard fine-tuning would have, which is likely because it is still preserving the original LM. One important baseline would be to train a small model fine-tuned on the target data and augment that with the original model, e.g., through logit ensembling, or weight merging (if the model sizes are the same). like how memory decoder is being incorporated currently. I think this is likely a competitive baseline for catastrophic forgetting (related work: [1], which is using a model soup). However, there is a chance memory decoder still has interesting properties that this baseline won’t have (related to the first bullet point).
- Question: One benefit of nonparametric LMs is that the data can be replaced without training, although it still requires the datastore & the nearest neighbor index to be updated. In the case of memory decoder, it requires building a nearest neighbor index AND training a new LM. How much additional overhead is training? I assume it depends on the training steps. If that cost is negligible, I think it is good to highlight; if not, a trade-off analysis would be helpful.
- Question: In Section 5.2, is each model fine-tuned individually on the training set of each benchmark and evaluated on its corresponding test set? I believe the very poor performance of full-param fine-tuning on some datasets like HYP, AGN, and Yahoo is quite suspicious – especially for AGN and Yahoo, as they are balanced and stable, unlikt HYP which is a tiny dataset with imbalanced labels. Understanding this result would be helpful.
- As mentioned earlier briefly, experiments are done on a small scale and evaluated on easy datasets by today’s standard, e.g., there are many published papers evaluated on these benchmarks with GPT-2 models in 2021-2022, but not recently. Also, many experiments are using PPL as a metric, e.g., domain adaptation serves as a primary motivation for the whole work, but is only doing PPL evals, which are less convincing.

Overall, I don’t see major flaws in the paper – it is a solid and well-executed paper, and includes experiments that the paper is supposed to have. However, I think the paper is not *exciting* enough, largely because it doesn’t fully explore some of the deeper questions it could have touched on, e.g., where the memory decoder sits in the spectrum of nonparametric LMs vs. fine-tuning (related to the first bullet point). Moreover, expanding experiments is likely to signify its impact, e.g., adding stronger models with more challenging benchmarks, adding downstream evals to domain adaptation experiments, and including the baseline mentioned in thes second bullet point.

[1] “Model soups: averaging weights of multiple fine-tuned models improves accuracy without increasing inference time” https://proceedings.mlr.press/v162/wortsman22a.html

---

> ### Author Rebuttal · Authors · 2025-07-31
>
> We sincerely thank the reviewer for the thoughtful and constructive feedback, and we are pleased that the reviewer found our work well-motivated, comprehensively evaluated, and clearly written.
>
> > Weakness 1: Positioning of Memory Decoder
>
> We appreciate the reviewer's insightful question about where Memory Decoder sits in the spectrum between non-parametric LMs and fine-tuning. This is indeed an important question that we are happy to address.
>
> Memory Decoder is **fundamentally a parametric method** that learns to mimic the behavior of non-parametric retrievers. Our key motivation is to compress the knowledge stored in large non-parametric datastores into a compact parametric model, thereby **combining the memorization capabilities of non-parametric methods with the efficiency and generalization abilities of parametric approaches**.
>
> We hypothesize that Memory Decoder can effectively learn long-tail knowledge (similar to non-parametric methods) while maintaining the coherence and reasoning capabilities of parametric models. Unlike non-parametric methods that rely on simple similarity matching and suffer from reasoning tasks (as noted in [1]), our parametric approach should preserve the model's ability to perform reasoning and maintain semantic coherence.
>
> To validate our hypothesis, we conducted case studies on WikiText-103 examining two key aspects (We compare label probability of different methods on the highlighted token, using tokens before as context):
>
> - Long-tail Knowledge Learning:
>
> | Sentence | Memory Decoder | kNN-LM | Base LM |
> |----------|---------------|---------|----------|
> | he starred alongside actors Mark Strong and Derek ***Jacobi***. | **68.94%** | 9.39% | 0.12% |
> | The launch of HMS Dreadnought in ***1906*** by the Royal Navy raised the stakes | **98.65%** | 40.62% | 1.57% |
>
> Memory Decoder assigns significantly higher probabilities to factual long-tail knowledge compared to the base LM, successfully capturing the memorization benefits of non-parametric methods.
>
> - Semantic Coherence and Basic Reasoning:
>
> | Sentence | Memory Decoder | kNN-LM | Base LM |
> |----------|---------------|---------|----------|
> | In 2000 Boulter had a guest-starring role ***on*** the television series The Bill. | 40.11% | 8.07% | **45.51%** |
> | the Royal Armoured Corps in Britain created three tank squadrons for special overseas operations, known as 'A', 'B' and '***C***' Special Service Squadrons. | 50.10% | 10.76% | **63.04%** |
>
> Memory Decoder maintains probabilities closer to the base LM rather than blindly following kNN-LM's lower probabilities, demonstrating its ability to preserve coherent language modeling capabilities.
>
> These results suggest that Memory Decoder successfully occupies a unique position: **it enhances memorization of domain-specific and long-tail knowledge like non-parametric methods, while maintaining the generalization and reasoning capabilities inherent to parametric models**. We further validate this hypothesis through extensive downstream evaluations of ***13 modern datasets*** (see response to Question 3 and the last section of response to reviewer pgYC for detail), where Memory Decoder consistently outperforms both non-parametric methods and standard fine-tuning approaches.
>
> > Weakness 2: Ablations on fine-tuned model
>
> We thank the reviewer for this important suggestion. We have indeed compared with this baseline in our ablation study (Table 5), where "CE Only" represents a small model fully fine-tuned on the domain corpus. Our results show that augmenting the base LLM with this fine-tuned model cannot achieve the same performance as Memory Decoder (4.69 vs 3.59), and we will highlight this observation more prominently in our revised manuscript.
>
> To further validate our approach, we conducted additional ablation studies on GPT-2 models, demonstrating that Memory Decoder provides substantial improvements over logit ensembling with a fine-tuned model:
>
> | Model | Base | +Finetune(small) | +MemDec(small) |
> |-------|------|------------------|----------------|
> | small | 24.89 | 15.95 | 13.36 |
> | medium | 18.29 | 14.26 | 12.25 |
> | large | 15.80 | 13.13 | 11.53 |
> | xl | 14.39 | 12.30 | 10.93 |
> | Avg | 18.34 | 13.91 | 12.01 |
>
> Memory Decoder **achieves an additional 1.9 perplexity improvement on average compared to the fine-tuned model baseline**, demonstrating that our novel training objective (combining distribution alignment with language modeling) enables superior memorization of domain knowledge beyond standard fine-tuning.
>
> Regarding Model Soup, we appreciate the suggestion but note that its main contribution focuses on "averaging weights of multiple models fine-tuned with different hyperparameter configurations." Our method has a fundamentally different objective—learning to mimic non-parametric retrieval behavior—and achieves much better generalizability across different model sizes and architectures. We leave the exploration of weight merging approaches as an interesting direction for future work.
>
> > Question 1: Trade-off analysis
>
> We thank the reviewer for this important question about training overhead. The reviewer is correct that non-parametric LMs offer data updates without retraining, though they still require rebuilding the datastore and kNN index.
>
> For Memory Decoder, our training budget is comparable to standard DAPT approaches—approximately 1 epoch of training a 7B model, which we consider a reasonable one-time cost for domain adaptation.
>
> We acknowledge this trade-off, but emphasize significant practical advantages:
>
> - Model-independent deployment: While **kNN-LM requires building model-specific datastores for each model**, a single Memory Decoder serves multiple models sharing the same tokenizer.
>
> - Inference efficiency: Memory Decoder incurs only 28% inference overhead vs kNN-LM's 117% (Figure 4). Additionally, kNN-LM requires 500GB memory for 100M tokens, while Memory Decoder compresses this into a 0.5B model.
>
> The one-time training cost is offset by dramatic improvements in inference efficiency, storage, and cross-model reusability—highly favorable for production scenarios with stable domain data. We will make this trade-off analysis more explicit in our revised manuscript.
>
> > Question 2: Understanding significant drop on AGN and Yahoo
>
> We appreciate the reviewer's question and would like to clarify that the downstream tasks in our paper follow zero-shot evaluation protocols from [2] and do not involve fine-tuning on individual datasets. We apologize for the confusion caused by the term "full-param fine-tuning"—this should be understood as Domain Adaptive Pre-training (DAPT) on the Wikitext-103 corpus. We will revise this terminology for clarity.
>
> Previous work has shown that DAPT can adversely affect a model's prompting ability [3]. We believe this effect is amplified in our evaluation due to our use of domain-conditional PMI (DCPMI) scoring. To investigate this, we compared standard language modeling (LM) scores with DCPMI scores:
>
> | Model | Yahoo(LM) | Yahoo(DCPMI) | HYP(LM) | HYP(DCPMI) | Avg |
> |-------|-----------|--------------|---------|------------|-----|
> | gpt2-small | 0.466 | 0.495 | 0.639 | 0.638 | 0.559 |
> | +DAPT | 0.429 | 0.244(-0.251) | 0.608 | 0.361(-0.277) | 0.410 |
> | gpt2-xl | 0.520 | 0.499 | 0.628 | 0.609 | 0.564 |
> | +DAPT | 0.490 | 0.491 | 0.624 | 0.618 | 0.556 |
>
> While direct LM evaluation shows modest drops, DCPMI scores drop significantly for GPT-2 small. We hypothesize that since we use domain-conditional scoring with fuzzy verbalizers (following [2]), and since words in Yahoo and AGN's label space (e.g., "politics," "technology") appear frequently in Wikitext-103, DAPT increases the domain probability for these terms, causing DCPMI scores to drop dramatically. The GPT-2 XL results show that larger models are less susceptible to this dataset bias, but the general trend holds. We will clarify this evaluation setup in our revised manuscript.
>
> > Question 3: Evaluation on modern datasets
>
> We conducted extensive evaluations on 13 real-world domain-specific tasks across all domains and large-scale reasoning tasks (NQ and HotpotQA). Due to limited space, detailed results are shown in the last section of our response to Reviewer pgYC.
>
> Key findings:
>
> - Domain-Specific Tasks: Memory Decoder consistently outperforms DAPT across all domains: biomedical (56.76 vs 43.03), financial (74.68 vs 71.17), and legal (42.99 vs 40.74). While DAPT hurts prompting ability [3], Memory Decoder preserves and enhances zero-shot/few-shot performance.
>
> - Memory-Intensive Reasoning: Unlike kNN-LM which hurts reasoning [1], Memory Decoder achieves substantial improvements: NQ (28.01 vs 24.00 for kNN-LM) and HotpotQA (27.72 vs 24.48 for kNN-LM).
>
> These results validate that Memory Decoder successfully combines memorization benefits with reasoning capabilities, addressing the limitations of both approaches on real-world tasks.
>
> > Conclusion
>
> We sincerely thank the reviewer for the constructive feedback. We have addressed the deeper questions about Memory Decoder's position in the spectrum between non-parametric LMs and fine-tuning through detailed case studies (Response to Weakness 1), expanded the suggested baseline comparisons (Response to Weakness 2), and evaluated on 13 modern datasets and challenging reasoning benchmarks (Response to Question 3). We hope these additions have addressed your concerns about the paper's depth and impact, and we will incorporate all these analyses into our revised manuscript.
>
> > References
>
> [1] Geng, Shangyi, Wenting Zhao, and Alexander M. Rush. "Great Memory, Shallow Reasoning: Limits of $ k $ NN-LMs." arXiv preprint arXiv:2408.11815 (2024).
>
> [2] Shi, Weijia, et al. "knn-prompt: Nearest neighbor zero-shot inference, 2022b." URL https://arxiv. org/abs/2205.13792 (2022).
>
> [3] Cheng, Daixuan, Shaohan Huang, and Furu Wei. "Adapting large language models via reading comprehension." The Twelfth International Conference on Learning Representations. 2023.

---

> > ### Comment · Reviewer_2Vi7 · 2025-08-04
> >
> > Thank you for your rebuttal. I increased my score from 3 to 4. I think the additional results that show the Memory Decoder is better at handling long tail knowledge even compared to the previous nonparametric model is very compelling. I also think, because the method is very simple and easy to use, it is likely to be adapted in practical scenarios quite easily. I still think experiments on a very small scale and primarily evaluating on PPL is the main limitation of this work, but also think this can depend on the available resources and shouldn't be the major reason for rejection.

---

> > > ### Author Response · Authors · 2025-08-05
> > >
> > > Dear reviewer,
> > >
> > > Thank you for increasing your score and for your thoughtful feedback throughout the review process. We greatly appreciate your recognition of the Memory Decoder's effectiveness in handling long-tail knowledge and its practical applicability due to its simplicity.
> > >
> > > We acknowledge your concern regarding the experimental scale. We are actively working to address this limitation by securing additional computational resources to conduct larger-scale experiments. We plan to include these results in our future work.
> > >
> > > Thank you again for your constructive comments, which have significantly strengthened our paper !

---

### Official Review · Reviewer_pgYC · 2025-07-01

**Clarity:** 3
**Significance:** 3
**Originality:** 3
**Rating:** 5
**Confidence:** 4

**Summary:**

The authors propose a plug-and-play memory module that enables domain adaptation for LLMs. The so-called memory decoder is designed to imitate retrieval-based methods like kNN-LM but without requiring non-parametric retrieval at inference time. The method is evaluated across various model families (e.g., Qwen, LLaMA) and tasks, demonstrating reduced perplexity and preserved downstream performance. The approach is positioned as a middle ground between DAPT and RAG.

**Questions:**

- How many runs were performed for the experiment in Table 2?
- How sensitive is the performance to the alpha value?
- Can you improve the formatting conventions used in the paper?

**Ethical Concerns:**

["NO or VERY MINOR ethics concerns only"]

**Final Justification:**

After carefully reading the authors’ response and other reviewers’ comments, I am raising my score to 5 (Accept). The authors have addressed my concerns, particularly around the interpolation factor and evaluation on additional datasets. The new results on benchmarks are convincing and strengthen the paper’s contributions. In general, I agree with Reviewer hqPH that the paper is well-written and presents a novel approach to domain adaptation.

**Limitations:**

Yes.

**Paper Formatting Concerns:**

Please use \citep{} or \citet{} correctly based on the context.

**Quality:**

2

**Strengths And Weaknesses:**

**Strengths:**

+ The paper is clearly written and well-organized. The method and experimental results are presented in a coherent way. Strengths and weaknesses of the methods have been elaborated.

+ The proposed method is novel and technically sound. The idea of mimicking non-parametric retrieval distributions through KL divergence during pretraining is innovative and addresses a real efficiency bottleneck in domain adaptation.

+ The contribution sounds practically significant. The Memory Decoder module demonstrates strong plug-and-play capability across multiple models sharing the same tokenizer. This flexibility, combined with notable reductions in inference latency compared to retrieval-based methods like kNN-LM, makes the approach attractive for deployment in resource-constrained or latency-sensitive settings.

**Weaknesses:**

- A more thorough factor analysis and clearer interpretability would strengthen the findings. For example, the interpolation weight alpha, which balances the base model and Memory Decoder outputs during inference, is fixed to 0.6 throughout all experiments without enough justification. It will be better if the paper explores how performance varies with different alpha values and whether the gains are robust to this choice.

- Some of the experimental results are not clearly presented. For example, Table 1 includes underlined values without explanation, leaving the readers puzzled about their significance. In Table 2, the best results are not consistently highlighted across all columns. The paper also makes strong claims, such as Memory Decoder outperforming full-parameter fine-tuning on GPT2-medium, without specifying whether these conclusions hold in general. Is this result statistically significant and generally true across all settings? The number of experimental runs and whether results are averaged or from single trials are not disclosed, which leaves the readers puzzled.

- Minor suggestions on formatting. The manuscript uses \cite{} instead of the appropriate \citep{}, leading to inconsistent in-text citations (e.g., “Author et al. [2020]”). This indicates a lack of attention to submission formatting standards and should be corrected.  The spacing between tables and their captions is inconsistent (e.g., Tables 1 and 2 have unusually small gaps, while Tables 3 and 4 appear properly spaced). Such inconsistencies detract from the professional appearance of the submission.

---

> ### Author Rebuttal · Authors · 2025-07-31
>
> Thank you for the thoughtful feedback. We greatly appreciate the reviewer's recognition of our novel approach, technical soundness, and practical significance.
>
> > Weakness 1 & Question 2: Alpha sensitivity analysis
>
> We apologize for the confusion in our description. To clarify: we use α=0.6 consistently for all domain adaptation experiments (Sections 5.3 and 5.4), while for Sections 5.1 and 5.2, we tune α on validation sets following established practices from prior work [1,2].
>
> To address your concern about sensitivity, we conducted comprehensive experiments across all 11 Qwen models on the law domain with different α values:
> | α | 0.4 | 0.45 | 0.5 | 0.55 | 0.6 | 0.65 | 0.7 | 0.75 | 0.8 |
> |---|---|---|---|---|---|---|---|---|---|
> | Avg PPL | 4.21 | 4.17 | 4.14 | 4.119 | 4.110 | 4.112 | 4.126 | 4.155 | 4.201 |
> | Relative to α=0.6 | 1.0244 | 1.0146 | 1.0073 | 1.0022 | 1.0000 | 1.0005 | 1.0039 | 1.0109 | 1.0221 |
>
> The results demonstrate remarkable robustness: performance varies by less than 2.5% across the entire α range (0.4-0.8), with optimal performance near α=0.6. This validates our choice and shows that **Memory Decoder is not overly sensitive to this hyperparameter**, making it practical for deployment without extensive tuning.
>
> For completeness, the specific α values used in other experiments were:
>
> - Wikitext-103: (0.8, 0.6, 0.55, 0.55) for (small, medium, large, xl) models respectively
> - Downstream tasks: α=0.3 for both Memory Decoder and kNN-LM following [2]
>
> The trend of smaller α for larger models aligns with intuition—stronger base models require less augmentation from the memory component. The general pattern still centers around α=0.6, confirming it as a robust default choice.
>
> > Weakness 2 & Question 1: Experimental clarity and additional results
>
> We apologize for the confusion in our presentation and thank the reviewer for helping us improve clarity.
>
> First, regarding Table 1, the underlined values represent the second-best results. We will clarify this in the revised manuscript to avoid confusion.
>
> To further validate our claim about Memory Decoder outperforming full-parameter fine-tuning, we conducted additional experiments with different Memory Decoder sizes:
>
> | | small | med | large | xl | Avg |
> |---|---|---|---|---|---|
> | Base | 24.89 | 18.29 | 15.80 | 14.39 | 18.34 |
> | DAPT | 15.47 | 12.78 | 11.10 | 10.16 | 12.38 |
> | +MemDec(small) 117M | 13.36 | 12.25 | 11.53 | 10.93 | 12.01 |
> | +MemDec(med) 345M | 12.08 | 11.59 | 10.92 | 10.43 | 11.26 |
> | +MemDec(large) 774M | **11.67** | **11.23** | **10.83** | **10.28** | **11.00** |
>
> These results confirm that our observation holds across different scales:
>
> - GPT2-medium + small MemDec (12.25) outperforms GPT2-medium DAPT (12.78)
> - GPT2-large + medium MemDec (10.92) outperforms GPT2-large DAPT (11.10)
> - GPT2-xl + large MemDec (10.28) slightly underperforms GPT2-xl DAPT (10.16) by only 1.2%
>
> This pattern demonstrates that Memory Decoder consistently achieves competitive or superior performance compared to full-parameter fine-tuning while maintaining the crucial advantage of being plug-and-play across all model sizes.
>
> Regarding experimental runs: For perplexity experiments, we performed single runs as the evaluation is deterministic. For downstream tasks in Table 2, we follow the domain-conditional PMI scoring method from [2], the result is also deterministic since it does not involve sampling-based generation, therefore we also only perform single runs. We will clarify this methodological detail in our revised manuscript to ensure transparency.
>
> > Weakness 3 & Question 3: Formatting improvements
>
> Thank you for your constructive feedback on formatting. We sincerely apologize for these oversights. The citation formatting issue arose from switching our bibliography style from IEEE to plainnat during preparation, which requires different citation commands.
>
> We will correct this in the camera-ready version by:
>
> - Replacing \cite{} with \citep{} for all references that do not function as grammatical elements in the sentence.
> - We will standardize the spacing between all tables and their captions to maintain consistency throughout the manuscript.
>
> We appreciate the reviewer bringing these formatting issues to our attention, as they help us improve the professional presentation of our work. All formatting corrections will be implemented in the final version.
>
> > **Evaluation on modern tasks**
>
> Following suggestions by other reviewers, we conducted extensive evaluations on **13 real-world domain-specific tasks** across all three domains and large-scale memory-intensive reasoning tasks.
>
> - Domain-Specific Downstream Tasks
>
> Following the evaluation framework from [3], we tested on large-scale in-context learning benchmarks across biomedicine, finance, and law domains. Our results demonstrate that Memory Decoder maintains strong performance in both zero-shot and few-shot settings, while DAPT shows significant performance degradation—consistent with findings in [3] that DAPT adversely affects prompting abilities.
>
> *Biomedical Domain:*
>
> | | ChemProt(13-shot) | MQP(4-shot) | PubmedQA(0-shot) | RCT(10-shot) | USMLE(0-shot) | Avg |
> |---|---|---|---|---|---|---|
> | Qwen2.5-7B | 24.4 | 83.44 | 63.7 | 70.1 | **36.92** | 55.71 |
> | +DAPT | 17.59 | 76.22 | 65.7 | 21.0 | 34.64 | 43.03 |
> | +Memory 0.5B | 24.4 | **84.09** | **64.4** | **74.06** | 36.84 | **56.76** |
>
>
> *Financial Domain:*
>
> | | FiQA_SA(5-shot) | FPB(5-shot) | Headline(5-shot) | NER(20-shot) | ConvFinQA(0-shot) | Avg |
> |---|---|---|---|---|---|---|
> | Qwen2.5-7B | 80.46 | 70.96 | 87.08 | 68.92 | 60.53 | 73.59 |
> | +DAPT | 75.59 | 66.39 | 86.03 | **69.32** | 58.52 | 71.17 |
> | +Memory 0.5B | **81.34** | **71.25** | **87.95** | 69.21 | **63.69** | **74.68** |
>
> *Legal Domain:*
>
> | | SCOTUS (micro)(0-shot) | SCOTUS (macro) | CaseHOLD (micro)(0-shot) | CaseHOLD (macro) | UNFAIR-ToS(4-shot) | Avg |
> |---|---|---|---|---|---|---|
> | Qwen2.5-7B | 26.66 | 17.90 | 35.92 | 35.93 | 87.05 | 40.69 |
> | +DAPT | 28.33 | 16.82 | 35.70 | 35.69 | **87.18** | 40.74 |
> | +Memory 0.5B | **31.66** | **21.05** | **37.58** | **37.59** | 87.05 | **42.99** |
>
> This **preservation of in-context learning capabilities while enhancing domain-specific performance** represents a significant advantage over traditional domain adaptation methods.
>
> - Knowledge-Intensive Reasoning Tasks
>
> Following [4], we evaluated on Natural Questions and HotpotQA. Notably, [4] found that kNN-LM actually hurts performance on these reasoning-intensive tasks, despite the answer strings existing in the non-parametric datastore.
>
> To address this challenge, we trained a 1B Memory Decoder on the same large heterogeneous unlabeled corpus (513M tokens) used in [4]:
>
> | Method | NQ | HotpotQA |
> |--------|------|----------|
> | Llama3-8B | 23.64 | 25.14 |
> | + kNN-LM | 24.00 (+0.36) | 24.48 (-0.66) |
> | + Memory Decoder (1B) | **28.01** (+4.37) | **27.72** (+2.58) |
>
> Memory Decoder provides substantial improvements on both benchmarks while maintaining strong reasoning capabilities, unlike kNN-LM which experiences performance degradation. This demonstrates our approach successfully **enhances memorization capabilities through learned retrieval patterns without compromising reasoning abilities**—validating our method's effectiveness on knowledge-intensive applications where both factual recall and reasoning are crucial.
>
> These comprehensive results confirm that Memory Decoder successfully scales to modern benchmarks.
>
> > References
>
> [1] Khandelwal, Urvashi, et al. "Generalization through memorization: Nearest neighbor language models." arXiv preprint arXiv:1911.00172 (2019).
>
> [2] Shi, Weijia, et al. "kNN-Prompt: Nearest Neighbor Zero-Shot Inference." arXiv preprint arXiv:2205.13792 (2022).
>
> [3] Cheng, Daixuan, Shaohan Huang, and Furu Wei. "Adapting large language models via reading comprehension." The Twelfth International Conference on Learning Representations. 2023.
>
> [4] Geng, Shangyi, Wenting Zhao, and Alexander M. Rush. "Great Memory, Shallow Reasoning: Limits of $ k $ NN-LMs." arXiv preprint arXiv:2408.11815 (2024).

---

> > ### Author Response · Authors · 2025-08-05
> >
> > Dear Reviewer,
> >
> > Thank you for your positive review and constructive feedback on our paper. We appreciate your acknowledgment of our rebuttal. Should you have any questions or concerns about our responses, we would be happy to address them.

---

> ### Comment · Reviewer_pgYC · 2025-08-06
>
> Dear authors,
>
> Sorry, we didn't realize that this year's NeurIPS setting is withholding the Final Justification at this Discussion phase. Below, please see the paragraph we posted two days ago:
>
> > After carefully reading the authors’ response and other reviewers’ comments, I am raising my score to 5 (Accept). The authors have addressed my concerns, particularly around the interpolation factor and evaluation on additional datasets. The new results on benchmarks are convincing and strengthen the paper’s contributions. In general, I agree with Reviewer hqPH that the paper is well-written and presents a novel approach to domain adaptation.

---

> > ### Author Response · Authors · 2025-08-08
> >
> > Dear reviewer,
> >
> > Thank you for your thoughtful reconsideration of our paper and for raising your score to Accept. We are glad to see that our response has successfully addressed your concerns. We really appreciate your positive feedback on the new benchmark results and your recognition of our novel approach to domain adaptation. We will incorporate all the additional experiments and clarifications in the final version of the paper.

---

### Official Review · Reviewer_mSKs · 2025-07-03

**Clarity:** 3
**Significance:** 2
**Originality:** 3
**Rating:** 4
**Confidence:** 3

**Summary:**

This paper introduces Memory Decoder (MemDec), a domain adaption method that aims to combine the benefits of Domain Adaptive Pretraining (DAPT) and kNN-based RAG while avoiding their drawbacks. The core idea of the method is to train a transformer decoder. that learns to emulate the output distributions of a non-parametric k-NN retriever. Once trained on a specific domain, the MemDec can be intergrated with any pretrained LLM that shares the same tokenizer. The experiments show that a small MemDec of 0.5B can significantly reduce the perplexity across various models from the Qwen and Llama families.

**Questions:**

Q1: Can you include evaluations on real-world domain-specific tasks (e.g., biomedical QA or legal reasoning) rather than relying mainly on perplexity and general NLP benchmarks?

Q2: Have you considered comparing against in-context learning baselines in domain settings?

Q3: Could the interpolation weight be adaptive rather than fixed globally? And could you analyze how the interpolation weight affect the performance?

**Ethical Concerns:**

["NO or VERY MINOR ethics concerns only"]

**Final Justification:**

The author has performed extensive additional experiments to address my concerns about lacking realistic evaluation and comparison with in context learning methods, and their results show impressive improvements compared with the baselines.

**Limitations:**

The author assumes full access to domain-specific corpora to build the k-NN datastore for MemDec training, but some domains may lack publicly available data or impose strict privacy constraints. The author may reflect on whether MemDec is feasible in low-resource or privacy-sensitive domains.

**Paper Formatting Concerns:**

None.

**Quality:**

2

**Strengths And Weaknesses:**

Strength
S1: The MemDec method distills the benefits of non-parametric retrieval into a small parametric model, providing an efficient alternative to traditional domain adaption. it also reduces the inference latency compared to kNN RAG method while avoiding the cost of full-parameter fine-tuning.

S2: The MemDec has great generalizability with the "plug-and-play" capability for models using the same tokenizer.

S3: MemDec achieves significant perplexity reduction across all tested models and domains.

Weakness
S1: Lack of realistic evaluation. The experiment is centered around perplexity and some downstream NLP tasks. While perplexity is a useful proxy for language modeling, it may not capture effectiveness in domain-specific tasks such as question answering or reasoning.

S2: While MemDec avoid retrieval overhead at inference time, it requires constructing large kNN datastore and performing kNN-based retrieval to generate supervision signal for the training of MemDec, which is very computationally expansive.

S3: There is no comparison to in-context learning baselines, which is a common baseline in modern domain adaption.

S4: The inference-time blending between the base LLM and Memory Decoder is controlled by a static interpolation weight α. However, different inputs, domains, or tasks may benefit from different levels of domain-specific influence, but the paper does not analyze the effects of this parameter or provide justification for the chose value (i.e., α = 0.6 for all tasks and models).

---

> ### Author Rebuttal · Authors · 2025-07-31
>
> We sincerely thank the reviewer for the thoughtful and constructive feedback. We greatly appreciate the reviewer's recognition of the efficiency gains and plug-and-play capabilities of Memory Decoder.
>
> > Weakness 1 & Question 1: Lack of realistic evaluation
>
> We appreciate this valuable suggestion and have conducted extensive evaluations on **13 real-world domain-specific tasks** across all three domains (biomedicine, finance, law) and large-scale memory-intensive reasoning tasks (Natural Questions and HotpotQA).
>
> Due to space constraints, we provide detailed results in **the last section of our response to Reviewer pgYC**. Our key findings demonstrate that Memory Decoder significantly outperforms baselines across diverse real-world applications:
>
> - *Domain-specific in-context learning*: Across 13 benchmarks spanning biomedical (ChemProt, MQP, PubmedQA, RCT, USMLE), financial (FiQA_SA, FPB, Headline, NER, ConvFinQA), and legal domains (SCOTUS, CaseHOLD, UNFAIR-ToS), Memory Decoder consistently improves performance while DAPT shows degradation, consistent with the findings from [2]. For instance, on finance tasks, DAPT drops average performance from 73.59 to 71.17, while Memory Decoder improves to 74.68.
>
> - *Knowledge-intensive reasoning*: On Natural Questions and HotpotQA, where traditional kNN-LM actually hurts performance [1], Memory Decoder achieves remarkable improvements of +4.37 and +2.58 points respectively. This demonstrates our method's unique ability to **combine the memorization benefits of non-parametric retrievers with the innate reasoning capabilities of LLMs**, addressing a critical limitation where retrieval-based methods typically degrade reasoning performance.
>
> These comprehensive evaluations confirm that Memory Decoder's effectiveness extends well beyond perplexity metrics to real-world domain-specific applications where both factual knowledge and reasoning capabilities are essential.
>
> > Weakness 2: Training overhead
>
> We appreciate the reviewer's attention to computational considerations. While Memory Decoder requires constructing a kNN datastore during training, this cost is significantly amortized compared to other methods:
>
> - *One-time vs. per-model*: Unlike DAPT requiring separate training for each model, Memory Decoder trains once for an entire model family sharing the same tokenizer. Training one 0.5B Memory Decoder is far more efficient than performing DAPT on multiple models (0.5B-72B).
>
> - *Model-agnostic datastore*: **Unlike kNN-LM which requires model-specific datastores** for each base model, our approach constructs the datastore only once. Traditional kNN-LM must rebuild datastores when switching models, while Memory Decoder eliminates this redundancy.
>
> - *Long-term efficiency*: Organizations train one Memory Decoder per domain and use it across their model ecosystem until major version updates (e.g., 9 months between Qwen2 and Qwen3). Section 5.4 demonstrates efficient cross-vocabulary adaptation with minimal training, further amortizing costs.
>
> The one-time training cost, acknowledged in our limitations, yields significant long-term savings compared to repeated DAPT or maintaining model-specific datastores for each deployment.
>
> > Weakness 3 & Question 2: In-context learning baselines
>
> Thank you for this important question about in-context learning comparisons. We'd like to clarify our method's positioning and provide additional experimental evidence:
>
> - *Memory Decoder's positioning*: Memory Decoder is designed to enhance learning from unlabeled domain corpora, which is why we primarily compare against DAPT throughout the paper. Our approach should be viewed as **orthogonal to methods like in-context learning and supervised fine-tuning**, rather than a replacement. The key advantage is that Memory Decoder can work synergistically with these methods.
>
> - *In-context learning experiments*: To address your specific concern, we conducted additional experiments comparing in-context learning performance. Our results from the domain-specific tasks already demonstrate that Memory Decoder outperforms both DAPT and base models in zero-shot and few-shot settings. We further conducted in-context learning ablations on the CaseHOLD legal reasoning task:
>
> | Method | CaseHOLD(micro) | CaseHOLD(macro) |
> |---|---|---|
> | Qwen2.5-7B (0-shot) | 35.92 | 35.93 |
> | +Memory 0.5B (0-shot) | 37.58 | 37.59 |
> | Qwen2.5-7B (4-shot) | 36.34 | 36.35 |
> | +Memory 0.5B (4-shot) | 37.95 | 37.95 |
> | Qwen2.5-7B (8-shot) | 36.34 | 36.35 |
> | +Memory 0.5B (8-shot) | **38.29** | **38.30** |
> | Qwen2.5-7B (16-shot) | 36.09 | 36.10 |
> | +Memory 0.5B (16-shot) | 37.53 | 37.53 |
>
> These results reveal several key insights:
>
> -  Zero-shot Memory Decoder outperforms all few-shot base models, demonstrating the effectiveness of our domain knowledge encoding.
> -  Memory Decoder continues to benefit from in-context examples, with performance improving from 0-shot to 8-shot settings, which shows our method is orthogonal to in-context learning methods.
> -  Even when both methods experience slight degradation at 16-shot (likely due to noise introduced by increased examples), Memory Decoder maintains its advantage over the base model.
>
> This demonstrates that Memory Decoder not only provides strong zero-shot domain adaptation but also **preserves and enhances the model's ability to leverage in-context examples**—a critical capability that DAPT often compromises.
>
> > Weakness 4 & Question 3: Adaptive interpolation weight
>
> We apologize for the confusion in our description. To clarify: we use α=0.6 consistently for all domain adaptation experiments (Sections 5.3 and 5.4), while for Sections 5.1 and 5.2, we tune α on validation sets following established practices from prior work [3,4].
>
> To address your concern about sensitivity, we conducted comprehensive experiments across all 11 Qwen models on the law domain with different α values:
>
> | α | 0.4 | 0.45 | 0.5 | 0.55 | 0.6 | 0.65 | 0.7 | 0.75 | 0.8 |
> |---|---|---|---|---|---|---|---|---|---|
> | Avg PPL | 4.21 | 4.17 | 4.14 | 4.119 | 4.110 | 4.112 | 4.126 | 4.155 | 4.201 |
> | Relative to α=0.6 | 1.0244 | 1.0146 | 1.0073 | 1.0022 | 1.0000 | 1.0005 | 1.0039 | 1.0109 | 1.0221 |
>
> The results demonstrate remarkable robustness: performance varies by less than 2.5% across the entire α range (0.4-0.8), with optimal performance near α=0.6. This validates our choice and shows that Memory Decoder is not overly sensitive to this hyperparameter, making it practical for deployment without extensive tuning.
>
> For completeness, the specific α values used in other experiments were:
>
> - Wikitext-103: (0.8, 0.6, 0.55, 0.55) for GPT2-(small, medium, large, xl) models respectively
> - Downstream tasks: α=0.3 for both Memory Decoder and kNN-LM following [3]
>
> The trend of smaller α for larger GPT models aligns with intuition—stronger base models require less augmentation from the memory component. The general pattern still centers around α=0.6, confirming it as a robust default choice.
>
> Regarding adaptive interpolation: As proven in [5], non-parametric LMs have very high ceiling if the interpolation weight is optimal. We explored training-free adaptive methods like those mentioned in [6] but observed minimal improvements. For simplicity, we don't include it in our final model. However, we note that the parametric nature of our method makes it particularly suitable for adaptive weights, which we leave for future work.
>
> > Limitation:
>
> Thank you for raising this important consideration about privacy-sensitive and low-resource domains. Memory Decoder actually offers significant advantages in these scenarios:
>
> - Privacy-preserving training: For privacy-sensitive domains, differential privacy techniques from [7] can be incorporated during kNN datastore construction. This adds calibrated noise to k-NN searches while maintaining utility, enabling Memory Decoder training on sensitive data without memorizing individual examples.
>
> - Enhanced deployment privacy: Crucially, Memory Decoder actually **offers superior privacy guarantees** compared to non-parametric methods. Unlike kNN-LM/RAG requiring corpus access at inference, Memory Decoder distills knowledge into parameters. Once trained, the original sensitive data can be completely removed from the deployment environment.
>
> - Low-resource advantages: Memory Decoder's efficiency is particularly valuable for low-resource domains. The one-time training cost is amortized across all models in a family, and the compact 0.5B model is far more practical to deploy than large datastores, making domain adaptation accessible in resource-constrained settings.
>
> These characteristics position Memory Decoder as an attractive solution for organizations dealing with sensitive or limited domain data.
>
> > References:
>
> [1] Geng, Shangyi, Wenting Zhao, and Alexander M. Rush. "Great Memory, Shallow Reasoning: Limits of $ k $ NN-LMs." arXiv preprint arXiv:2408.11815 (2024).
>
> [2] Cheng, Daixuan, Shaohan Huang, and Furu Wei. "Adapting large language models via reading comprehension." The Twelfth International Conference on Learning Representations. 2023.
>
> [3] Shi, Weijia, et al. "knn-prompt: Nearest neighbor zero-shot inference, 2022b." URL https://arxiv. org/abs/2205.13792 (2022).
>
> [4] Khandelwal, Urvashi, et al. "Generalization through memorization: Nearest neighbor language models." arXiv preprint arXiv:1911.00172 (2019).
>
> [5] Xu, Frank F., Uri Alon, and Graham Neubig. "Why do nearest neighbor language models work?." International Conference on Machine Learning. PMLR, 2023.
>
> [6] Li, Minghan, et al. "Nearest neighbor speculative decoding for llm generation and attribution." Advances in Neural Information Processing Systems 37 (2024): 80987-81015.
>
> [7] Zhu, Yuqing, et al. "Private-knn: Practical differential privacy for computer vision." proceedings of the IEEE/CVF conference on computer vision and pattern recognition. 2020.

---

> > ### Author Response · Authors · 2025-08-05
> >
> > Dear Reviewer,
> >
> > We hope you've had a chance to review our detailed response to your concerns. We would greatly appreciate your feedback on whether our clarifications and additional experimental results have addressed the issues you raised, or if there are any remaining concerns we can address.

---

> > ### Comment · Reviewer_mSKs · 2025-08-07
> >
> > Thanks for the authors' detailed and thoughtful response to my questions, the additional evaluation on the domain-specific tasks have dismissed much of my concern about the paper, and I have updated my score accordingly. I suggest the authors to include these experiments in their final version of the paper.

---

> > > ### Author Response · Authors · 2025-08-08
> > >
> > > Dear reviewer,
> > >
> > > Thank you for your careful consideration of our response and for providing instructive feedback on our paper. We are glad to see that our rebuttal has resolved much of your concerns. We will include all additional experiments in the final version of the paper as suggested.

---

### Official Review · Reviewer_hqPH · 2025-07-05

**Clarity:** 3
**Significance:** 3
**Originality:** 3
**Rating:** 5
**Confidence:** 4

**Summary:**

The authors present an approach to augmenting language models with non-parametric/neural adapters that are trained on domain specific data. The training procedure constructs pairs based on a kNN search to find relevant documents based on hidden representations of the pre-trained model. The pre-training objective aligns the memory decoder's output distribution and the kNN distributions for each sample; the overall loss an interpolation of the KL and LM losses. The authors show improved average performance on in-context learning and domain adaptation benchmarks in context, along with ablation studies on the pre-training objective, along with the setup's ability to adapt across different vocabularies.

**Questions:**

- What is the computational cost and dataset size of the pre-training KNN dataset? How many samples are needed for training? A brief mention of the cost of this for large datasets (even if it pales in comparison to costs incurred during inference) would help.
- How much does the size of the transformer decoder storing memory affect performance for in-context tasks? Ablations would help elucidate where some of the in-context learning gains are coming from.
- What's the detailed setup for the full parameter finetuning baseline? There's a width breadth of finetuning approaches with varying levels of effectiveness beyond simply retraining the whole model on the domain-specific corpus.
- Consider also citing [Eyuboglu et al. 2025](https://arxiv.org/abs/2506.06266) in recent work along with other architectures around memory-based augmentations to architectures such as [Zhang et al.](https://arxiv.org/pdf/2405.06394).
- What are the capacity limitations of the decoder based on model size? Does performance show degradation with larger-sized in-context learning benchmarks?

**Ethical Concerns:**

["NO or VERY MINOR ethics concerns only"]

**Final Justification:**

See my recent responses. The adjustment in my score is due to stolid results from larger in-context learning benchmarks, the ablation studies demonstrating the efficiency of the approach, and the results on other QA tasks being promising.

**Quality:**

3

**Strengths And Weaknesses:**

Strengths:
- The presented approach is novel, simple, and not particularly resource intensive to train.
- The manuscript is well-written and easily to understand.
- The proposed approach shows results that broadly improve performance and give balanced accuracy across relevant benchmarks.
- The authors ablate some of their proposed approach, including the choice of loss function.

Weaknesses:
- Ablations top of the decoder architecture itself would elucidate more about the particular attributes of the proposed approach that improve overall performance; this would strengthen the credibility of the approach and give some predictive intuition around how it might scale.
- The scale of the experiments is not particularly large per larger-scale in-context learning tasks. The authors might consider evaluating on tasks such as [Arora et al.](https://direct.mit.edu/tacl/article/doi/10.1162/tacl_a_00580/117168) or on other larger-scale QA benchmarks such as [Yang et al.](https://arxiv.org/abs/1809.09600) that might require larger-scale domain knowledge.

---

> ### Author Rebuttal · Authors · 2025-07-31
>
> Thank you for the thoughtful feedback. We greatly appreciate the reviewer's recognition of our novel approach and strong empirical results.
>
> > Weakness 1: Architecture choice
>
> We appreciate the reviewer's suggestion for architecture ablations. Our choice of transformer decoder was driven by **our training objective: learning to predict kNN distributions from context tokens autoregressively**. Transformer decoders have proven exceptional at modeling complex probability distributions, which is precisely what we need for mimicking retriever behavior.
>
> Our results validate this choice—with just 117M parameters, our decoder successfully captures retrieval patterns that would otherwise require searching through massive datastores (e.g., 500GB for Wikitext-103). While we acknowledge that other architectures like specialized memory networks might also be well-suited for distribution imitation, we focused this initial work on demonstrating that pre-trained memory decoders are viable and effective across diverse domains and model scales. We view architectural exploration as a valuable direction for future work.
>
> > Weakness 2: Performance on Large-scale QA benchmarks
>
> We thank the reviewer for this excellent suggestion. Following the reviewer's recommendation, we conducted additional large-scale experiments on knowledge-intensive tasks, specifically evaluating on Natural Questions (NQ) and HotpotQA benchmarks. We closely followed the experimental setup from [1] to ensure fair comparison.
>
> Interestingly, [1] found that kNN-LM actually **hurts** performance on these reasoning-intensive tasks, despite the answer strings existing in the non-parametric datastore.
>
> To address this challenge, we trained a 1B Memory Decoder on the same **large heterogeneous unlabeled corpus (513M tokens)** used in the previous study. Our results demonstrate that Memory Decoder successfully overcomes the limitations of traditional retrieval methods:
>
> | Method | NQ | HotpotQA |
> |--------|------|----------|
> | Llama3-8B | 23.64 | 25.14 |
> | + kNN-LM | 24.00 (+0.36) | 24.48 (-0.66) |
> | + Memory Decoder (1B) | **28.01** (+4.37) | **27.72** (+2.58) |
>
> We show that **Memory Decoder achieves the best of both worlds**: it provides substantial improvements on both benchmarks (+4.37 on NQ, +2.58 on HotpotQA) while avoiding the reasoning degradation observed with kNN-LM. This demonstrates that our approach successfully enhances memorization capabilities through learned retrieval patterns without compromising the model's reasoning abilities—a key limitation of traditional retrieval methods.
>
> > Question 1: Computational Cost and Dataset size
>
> We appreciate the reviewer's question about training costs. Computationally, Memory Decoder requires FLOPs equivalent to 1 epoch of training a 7B model—negligible compared to DAPT needed for each model from 0.5B to 72B.
>
> For dataset size, we used ~100M tokens for specialized domains and successfully scaled to **500M tokens** for a large heterogeneous corpus (Wikipedia + CC-NEWS) in response to Weakness 2, demonstrating effective handling of larger datasets.
>
> While experiments beyond 500M tokens would provide additional insights, our current evaluation comprehensively demonstrates Memory Decoder's effectiveness across diverse domains and scales. This work establishes a fundamentally novel approach—training plug-and-play memory components to imitate retrieval behavior—opening new research directions in domain adaptation. We view larger-scale exploration as valuable future work as this new paradigm matures.
>
> > Question 2: Ablation on the size of Memory Decoder
>
> We appreciate the reviewer's question and would like to clarify that the downstream tasks in our paper follow zero-shot evaluation protocols from [2]. Our Memory Decoder is designed to **enhance learning from unlabeled domain corpora**, which is why we primarily compare against DAPT throughout the paper. Our approach should be viewed as orthogonal to methods like in-context learning and supervised fine-tuning. For additional experiments on in-context learning capabilities, please see our response to Question 5.
>
> To address the reviewer's suggestion, we conducted ablations on Memory Decoder size:
>
> | Model | GPT2-small | GPT2-med | GPT2-large | GPT2-xl | Avg |
> |-------|------------|----------|------------|---------|------|
> | Base | 24.89 | 18.29 | 15.80 | 14.39 | 18.34 |
> | DAPT | 15.47 | 12.78 | 11.10 | 10.16 | 12.38 |
> | +MemDec(117M)  | 13.36 | 12.25 | 11.53 | 10.93 | 12.01 |
> | +MemDec(345M)  | 12.08 | 11.59 | 10.92 | 10.43 | 11.26 |
> | +MemDec(774M)  | **11.67** | **11.23** | **10.83** | **10.28** | **11.00** |
>
> These results reveal that our small Memory Decoder (117M) already achieves excellent performance across all model sizes. While larger decoders provide incremental improvements, the additional gains do not justify the increase in computational cost.
>
> This finding validates our design choice of using compact Memory Decoders, demonstrating that effective domain adaptation can be achieved without scaling to larger decoder sizes. The efficiency of our approach makes it practical for real-world deployment.
>
> > Question 3: Setup for DAPT
>
> We appreciate the reviewer's question about our baseline setup. For full-parameter finetuning, we follow the standard Domain Adaptive Pre-Training (DAPT) approach from [3]. Specifically, we continue full-parameter training on the domain-specific corpus for 1 epoch using a learning rate of 1e-5.
>
> While we acknowledge that various DAPT approaches exist with different training strategies like [4], we chose the standard DAPT setup as it remains the most prevalent and widely-adopted method in domain adaptation literature. This baseline provides a well-established benchmark that enables meaningful comparison and validates the effectiveness of Memory Decoder within the broader domain adaptation landscape. Our consistent improvements over this standard approach across multiple domains and model scales demonstrate the practical advantages of our plug-and-play memory paradigm.
>
> > Question 4: Additional references
>
> We thank the reviewer for bringing these relevant works to our attention. We will include these citations in the camera-ready version.
>
> After reviewing these papers, we do note that these works address different problems at smaller scales. Eyuboglu et al. (2025) focuses on task-specific memory augmentation for reasoning, while Zhang et al. explores memory mechanisms mainly on a small synthetic dataset. In contrast, Memory Decoder tackles domain adaptation at scale, enhancing up to 72B parameters across entire domains (500M+ tokens).
>
> Nevertheless, these works provide valuable insights, particularly their architectural innovations, that could inform future Memory Decoder designs. We appreciate the reviewer highlighting these connections.
>
> > Question 5: Capacity limitations
>
> We appreciate the reviewer's thoughtful question about capacity limitations and performance on larger benchmarks.
>
> - Regarding capacity limitations:
>
> Recent theoretical work [5] provides insight into memorization capacity in language models. According to [5], models can store approximately 3.6 bits per parameter when completely eliminating generalization. Given that English text contains 1-2 bits of information per character [6] and following the general approximation of 1 token ≈ 4 characters, **we can estimate that complete memorization of the Wikitext-103 dataset would theoretically require at least 200M parameters**.
>
> Remarkably, our Memory Decoder achieves exceptional memorization and generalization with only 117M parameters on Wikitext-103, suggesting that our proposed paradigm exhibits novel scaling properties distinct from traditional language models. This efficiency indicates the existence of new scaling laws specific to our training objective, presenting exciting opportunities for future theoretical exploration.
>
> - Regarding performance on Large-Scale Benchmarks:
>
> Apart from large-scale QA datasets that we showed in our response to Weakness 2, we also conducted extensive evaluations on **13 large-scale in-context learning benchmarks** across all three domains. Due to length constraints, we provide detailed results for all datasets in the last section of our response to Reviewer pgYC.
>
> In summary, across 13 benchmarks spanning biomedical (ChemProt, MQP, PubmedQA, RCT, USMLE), financial (FiQA_SA, FPB, Headline, NER, ConvFinQA), and legal domains (SCOTUS, CaseHOLD, UNFAIR-ToS), Memory Decoder consistently improves performance while DAPT shows degradation in both zero-shot and few-shot settings, consistent with the findings from [4]. For instance, on finance tasks, DAPT drops average performance from 73.59 to 71.17, while Memory Decoder improves to 74.68. This demonstrates our approach successfully scales to large-sized in-context learning benchmarks.
>
> > References
>
> [1] Geng, Shangyi, Wenting Zhao, and Alexander M. Rush. "Great Memory, Shallow Reasoning: Limits of $ k $ NN-LMs." arXiv preprint arXiv:2408.11815 (2024).
>
> [2] Shi, Weijia, et al. "knn-prompt: Nearest neighbor zero-shot inference, 2022b." URL https://arxiv. org/abs/2205.13792 (2022).
>
> [3] Gururangan, Suchin, et al. "Don't stop pretraining: Adapt language models to domains and tasks." arXiv preprint arXiv:2004.10964 (2020).
>
> [4] Cheng, Daixuan, Shaohan Huang, and Furu Wei. "Adapting large language models via reading comprehension." The Twelfth International Conference on Learning Representations. 2023.
>
> [5] Morris, John X., et al. "How much do language models memorize?." arXiv preprint arXiv:2505.24832 (2025).
>
> [6] Shannon, Claude E. "Prediction and entropy of printed English." Bell system technical journal 30.1 (1951): 50-64.

---

> > ### Author Response · Authors · 2025-08-05
> >
> > Dear Reviewer,
> >
> > Thank you for your positive review of our work. We hope you've had the opportunity to review our rebuttal and additional results. We would appreciate any feedback you might have on our responses.

---

> > > ### Author Response · Authors · 2025-08-08
> > >
> > > Dear reviewer,
> > >
> > > We just want to gently remind you that the rebuttal deadline is in less than two days. We are wondering if our response has adequately addressed your concerns and would be happy to clarify or discuss any remaining questions you might have.

---

> > > > ### Comment · Reviewer_hqPH · 2025-08-08
> > > >
> > > > I appreciate the responses from the authors:
> > > > - the HotpotQA results are helpful to validate the approach on more complex QA tasks.
> > > > - the larger in-context learning benchmarks are additionally helpful to validate generalization
> > > > - thank you for the answers regarding dataset size
> > > > - the ablations on memory decoder size show that the approach is efficient insofar as the returns diminish with larger models, which, for the scope and aim of the work, is coherent.
> > > >
> > > > Given the responses, I am adjusting my score to 5.

---

> > > > > ### Author Response · Authors · 2025-08-08
> > > > >
> > > > > Dear reviewer,
> > > > >
> > > > > Thank you for your careful consideration of our responses. We are glad to see that the HotpotQA results and larger in-context learning benchmarks have helped validate our approach's generalization capabilities. We will include all these additional experiments and analyses in the final version of the paper.

---

### Decision · Program_Chairs · 2025-09-17

**Decision:**

Accept (poster)

**Comment:**

The work proposes memory decoder, a small pretrained decoder that can adapt LLMs to specialized domains without domain specific pre-training or RAG. This is achieved by training a memory module to imitate the output the distribution of a k-NN retriever. This is a well motivated work solving an important problem. The proposed approach is novel and shown to be effective based on empirical studies. The "plug-and-play" nature of the proposal makes the work quite practical.

The authors' rebuttal addressed most of the concerns, and all reviewers recommended acceptance after the rebuttal and discussion period. Overall, the AC finds the work to be a good contribution to the community. Please incorporate the new experiments and discussions in the final version.

One question from 2Vi7 worths further discussions in the final version:

"a nonparametric LM and finetuning of LM have different properties, rather than one being universally better than the other, e.g., a nonparametric model may be better at long-tail or uncommon patterns like factoid knowledge, while finetuning of an LM may make the model generalize better. Is the proposed model closer to a nonparametric LM, or to finetuning of LM, in terms of model properties?"

The authors' reply and additional experiments in the rebuttal show that "Memory Decoder can effectively learn long-tail knowledge (similar to non-parametric methods) while maintaining the coherence and reasoning capabilities of parametric models. "

However it is easy to construct an external knowledge set such that the "long-tail" knowledge cannot be fully compressed/memorized by a small LM. For example, in the extreme case, one can construct a table (as external knowledge) mapping a set of keys to a set of (random) values, and the size of the table is larger than the size of the small memory decoder. In such a case, it is not possible to compress the knowledge into the memory decoder.

The authors should tone down the statement a bit and discuss the limitations.